# Hippocampome.org: a knowledge base of neuron types in the rodent hippocampus

Diek W Wheeler, Charise M White, Christopher L Rees, Alexander O Komendantov, David J Hamilton, Giorgio A Ascoli*

Krasnow Institute for Advanced Study, George Mason University, Fairfax, United States

**Abstract** Hippocampome.org is a comprehensive knowledge base of neuron types in the rodent hippocampal formation (dentate gyrus, CA3, CA2, CA1, subiculum, and entorhinal cortex). Although the hippocampal literature is remarkably information-rich, neuron properties are often reported with incompletely defined and notoriously inconsistent terminology, creating a formidable challenge for data integration. Our extensive literature mining and data reconciliation identified 122 neuron types based on neurotransmitter, axonal and dendritic patterns, synaptic specificity, electrophysiology, and molecular biomarkers. All ~3700 annotated properties are individually supported by specific evidence (~14,000 pieces) in peer-reviewed publications. Systematic analysis of this unprecedented amount of machine-readable information reveals novel correlations among neuron types and properties, the potential connectivity of the full hippocampal circuitry, and outstanding knowledge gaps. User-friendly browsing and online querying of Hippocampome.org may aid design and interpretation of both experiments and simulations. This powerful, simple, and extensible neuron classification endeavor is unique in its detail, utility, and completeness.

## Introduction

A century after the advent of the neuron doctrine (*Jones, 2006*), neuroscience is challenged with data on neuronal characteristics spread throughout thousands of articles in hundreds of journals, growing at a rate of dozens of new papers each day. The sweeping array of research designs, data formats, and presentation styles, reflecting the tremendous diversity in neuronal structure and functions, makes it extremely difficult to assess the available information. Overcoming the hindrances of this variety is problematic because of limited consensus on which properties to use for neuron classification and how to define them effectively (*Petilla Interneuron Nomenclature Group et al., 2008*; *Hamilton et al., 2012*). The lack of a complete accounting of neuron types is recognized as a critical omission for experimental and computational progress alike (*Lichtman and Denk, 2011*).

As a case in point, the rodent hippocampus is among the most intensively studied neural systems. Numerous seminal efforts have summarized the enormous amount of information on the morphology, connectivity, biochemistry, and electrophysiology of hippocampal neurons (*Bernard and Wheal, 1994*; *Patton and McNaughton, 1995*; *Freund and Buzsáki, 1996*; *Parra et al., 1998*; *McBain and Fisahn, 2001*; *Maccaferri and Lacaille, 2003*; *Amaral and Lavenex, 2007*; *Canto et al., 2008*; *Klausberger and Somogyi, 2008*; *Klausberger, 2009*; *Somogyi, 2010*; *Bezaire and Soltesz, 2013*). However, these notable advances have yet to translate into an integrated understanding of corresponding functions. The hippocampus plays a critical role in the consolidation and retrieval of episodic memory (*Nadel and Moscovitch, 1997*; *Wang and Morris, 2010*) as well as in spatial representation and navigation (*Foster and Knierim, 2012*; *Moser et al., 2015*). Many theories have been formulated to connect these important cognitive functions to hippocampal architecture (*Clark and Squire, 2010*; *Eichenbaum and Cohen, 2014*), rhythmic activity (*Burgess and O'Keefe, 2011*;

*For correspondence: ascoli@gmu.edu

**Competing interests:** The authors declare that no competing interests exist.

**eLife digest** The hippocampus is a seahorse-shaped region of the brain that is responsible for learning, emotions, and memory. Like other regions of the brain, it contains many types of neurons that send information to each other by releasing chemicals called neurotransmitters across junctions known as synapses. Identifying all the different neuron types in the hippocampus is an important step towards understanding in detail how this brain region works.

Thousands of articles have been published that attempt to characterize the neurons in the hippocampus, but many of these studies report only some of the properties of a new neuron type. It is also often difficult to compare the results of different studies, as many different approaches have been used to investigate neuron types, and different studies often use different terms to describe similar features.

Wheeler et al. have now created a resource called Hippocampome.org that combines approximately 14,000 pieces of experimental evidence about neuron types in the rat hippocampus into a unified database. Analyzing these data has revealed about 3700 different neuron properties. By primarily considering the pattern formed by the branched axons and dendrites, the outputs and inputs that extend out of a neuron, Wheeler et al. have identified over a hundred different neuron types. This classification system also considers how selectively the neuron forms synapses with other cells and the identity of the neurotransmitter released by the neuron. In the future, other features of the neurons will also be incorporated into the system to help refine the classifications.

All of this information is online and freely available at Hippocampome.org. This resource is expected to provide a solid basis for analyzing how the hippocampus works, by helping researchers to design and interpret experiments and simulations.

*Hasselmo and Stern, 2014*), and synaptic plasticity (*Rolls and Treves, 1994*; *Bliss and Collingridge, 2013*). However, current models are far from comprehensively accounting for the available experimental data on all neuron types and relevant properties.

As a stepping stone towards filling this gap, we mounted a large-scale literature mining effort to assemble a knowledge base of neuron types in the rodent hippocampal formation: dentate gyrus (DG), CA3, CA2, CA1, subiculum (Sub), and entorhinal cortex (EC). Our data-driven analysis of thousands of peer-reviewed publications identified a specific set of neuron properties suitable to define a basic classification scheme for collating, organizing, and integrating available knowledge. A cornerstone of this approach is the distributions of axons and dendrites across the distinct areas and layers of the hippocampal formation, such as CA1 stratum radiatum (SR) or EC layer II. Axonal and dendritic arbors perennially have been central to the experimental identification of neurons (*Jones, 2006*; *DeFelipe et al., 2013*), as they underlie network connectivity and profoundly influence information processing. Systematic mining of published data on axonal and dendritic profiles, augmented with information on neurotransmitter and synaptic specificity, led to the tentative definition of over 100 distinct neuron types across the hippocampal formation. Importantly, this initial identification of 'morphological' types enabled the unambiguous incorporation of existing molecular and electrophysiological data into the knowledge base. Other key features, such as developmental origin, firing dynamics, synaptic properties, neuron counts, and connectivity ratios, are being progressively integrated into the same classification framework.

Essential to this effort is the digestion of information from original publications into both human- and machine-readable forms. We present Hippocampome.org, a publicly accessible web portal to browse and query data with direct links to specific evidence in the scientific literature. Information summaries are available for each neuron type, anatomical parcel, molecular marker, and cited author. Searching for combined morphological, molecular, and electrophysiological properties returns lists of all known neuron types with those features. These functionalities, coupled with the knowledge base's comprehensiveness and inter-relatedness, reveal novel insights on neuron types, their properties, and circuit connectivity, which are impractical or impossible to derive from traditional literature searches. Moreover, the collation of data enables quantification of available information, unearthing residual knowledge gaps.

The 'Description of resource' section of this article aims to offer a clear description of both the conceptual foundation and practical utility of Hippocampome.org according to the following

organization. First, we explain the neuron type classification criteria, including the definition of morphological patterns and the distinctions by main neurotransmitter, synaptic specificity, biomarkers, and electrophysiology. We then provide a summary of the knowledge base content and describe its online accessibility. Next, we illustrate the usefulness and possible applications of Hippocampome.org through analyses of pairwise correlations among neuron properties, potential circuit connectivity, and use case scenarios in experimental and computational investigations. A discussion follows the 'Description of resource' section. In the final 'Materials and methods' section, we provide practical details of how the Hippocampome.org knowledge base was assembled. We define parcels and neuron types, explain how biomarker and electrophysiological data are linked to morphological data, describe how names are assigned to neuron types, and expand upon how the knowledge base will be maintained going forward. For optimal comprehension of this resource, we recommend online consultation of Hippocampome.org while reading this article.

## Description of resource

### Identifying neuron types by axonal and dendritic patterns

As a first step in the knowledge-base design, we sought the optimal level of description to capture the largest possible extent of available information. For example, just distinguishing neurons into projection cells and local interneurons is too coarse to reflect the known variety of hippocampal neuron types. In contrast, demanding a detailed quantification of axonal and dendritic morphology and connectivity excludes the majority of available scientific reports that only depict neuronal arbors qualitatively. An intermediate approach is to describe axons and dendrites based on the specific areas and layers they invade. The orderly anatomical organization of the hippocampal formation is commonly delineated into 26 cytoarchitectural parcels (*Figure 1A,B*): DG outer stratum (s.) moleculare (SMo), inner s. moleculare (SMi), s. granulosum (SG), hilus (H); CA3 s. lacunosum-moleculare (SLM), s. radiatum (SR), s. lucidum (SL), s. pyramidale (SP), s. oriens (SO); 4 each in CA2 and CA1 (same as CA3 except SL); Sub s. moleculare (SM), SP, polymorphic layer (PL); and EC layers I–VI. Most publications that report morphological information on hippocampal neurons include evidence of axonal and dendritic presence in at least a subset of these parcels in the form of reconstructions, tracings, microscopic images, schematics, or text descriptions (*Figure 1C–E*).

This relatively simple description is surprisingly effective to map neuronal diversity in the hippocampus. Specifically, the binary representation (present or not) of both axons and dendrites across the 26 hippocampal parcels (see 'Materials and methods') is sufficient to identify >100 unique morphological profiles based on existing literature (selection in *Figure 1F*; complete data: hippocampome.org/morphology). A key assumption of this framework is that neurons differing in axonal or dendritic patterns are different types. The same morphological pattern is further separated into distinct types if the neurons can be discriminated by neurotransmitter, synaptic selectivity, or consistent molecular and electrophysiological differences (see below), giving rise to 122 neuron types distinguished so far (18 in DG, 25 in CA3, 5 in CA2, 40 in CA1, 3 in Sub, and 31 in EC). The soma layer location is annotated for all neuron types, but neurons are not considered of separate types solely due to this feature if their other main characteristics are the same. Several neuron types have somata distributed across both sides of, or just along, a layer boundary, such as CA1 quadrilaminar cells (*Pawelzik et al., 2002*) or DG hilar commissural-associational pathway (HICAP) cells (*Mott et al., 1997*).

If only dendrites or axons are observed (but not both), or a single instance of a neuron morphology is described (n = 1), the characterization is deemed insufficient to include a neuron type in the knowledge base. We often attempt to contact authors to ascertain whether additional data may exist. If no further evidence is available, the existing information is extracted but not integrated with the rest of the knowledge base, and the possible neuron type is placed 'on hold' until more data are published. Currently, 151 potential neuron types are in this state (hippocampome.org/on-hold).

Hippocampome.org links information from various publications to specific neuron types primarily based on parcel-delineated neurite morphology and does not rely on author given names for neuron classification. The reason is that the same name is frequently employed in the literature to describe neurons with different morphological patterns. For example, the term 'bistratified' has been used to describe different morphologies corresponding to Hippocampome cell types CA1 Bistratified (*Freund and Buzsáki, 1996*), CA1 Radial Trilaminar (*Daw et al., 2009*), and Schaffer Collateral-receiving Radiatum-targeting (*Leão et al., 2012*). Furthermore, neurons of the same type are often

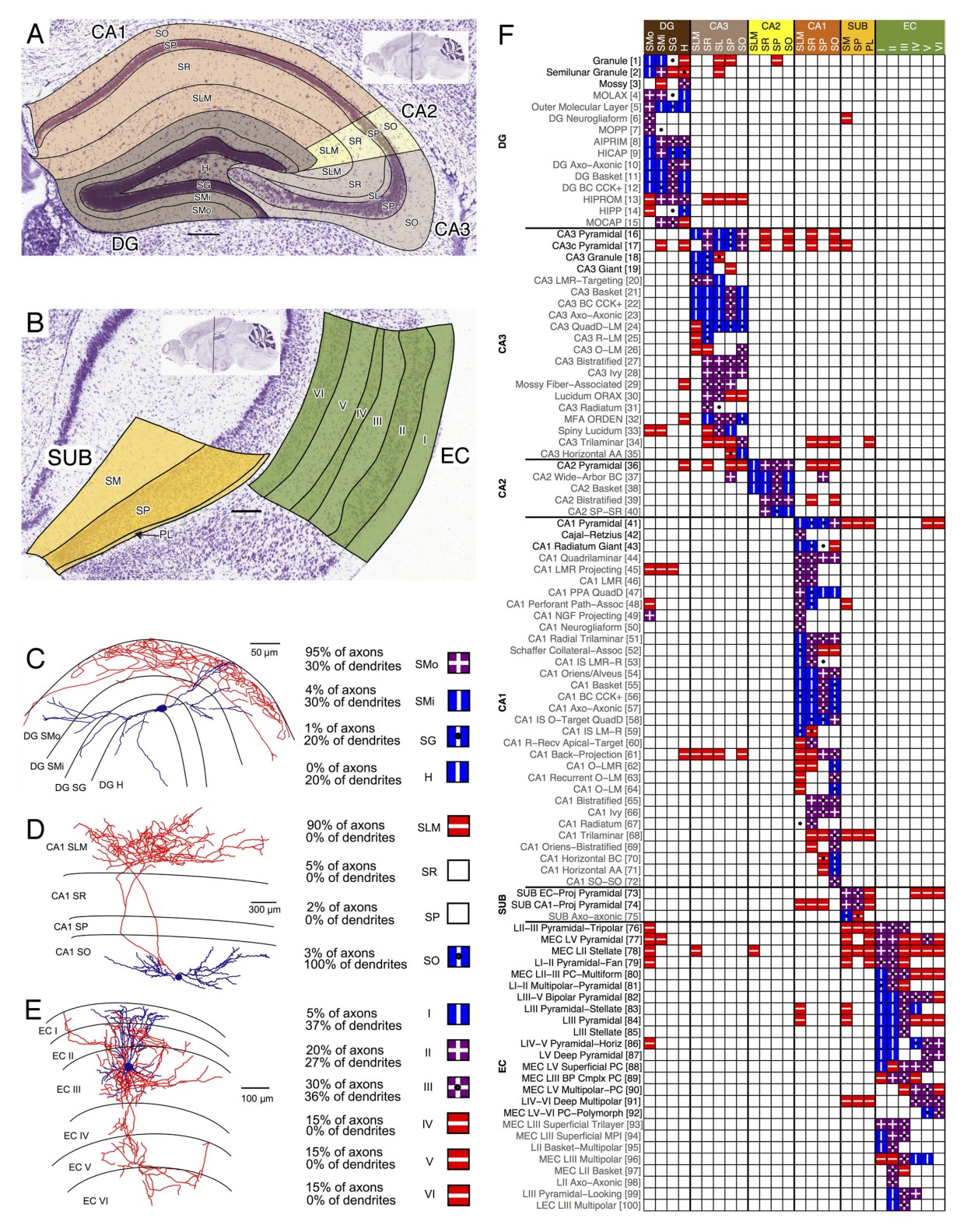

**Figure 1.** Defining neuron types with anatomical parcels and morphological patterns. (A, B) Nissl staining of a P56 mouse (coronal sections 74 and 85 from the Allen Brain Atlas) overlaid by color-coded parcels (in accord with Temporal-Lobe.com) of (A) the hippocampus proper and (B) the rest of the hippocampal formation. See main text for abbreviations. (C–E) Example morphological reconstructions from NeuroMorpho.Org (left) with red axons and blue dendrites; their estimated axonal and dendritic breakdown by layers (middle); and Hippocampome.org representation (right) with blue square and

*Figure 1. continued on next page*

*Figure 1. Continued*

vertical line (I) indicating dendritic presence, red square and horizontal line (—) indicating axonal presence, purple square and cross (+) indicating both axonal and dendritic presence, and a black dot (•) indicating soma location. (**C**) Dentate gyrus (DG) Outer Molecular Layer cell with axons in SMo and dendrites in all layers (NMO_00179; [*Mott et al., 1997*]). (**D**) CA1 O-LM cell with axons in SLM and dendrites in SO (NMO_02347; [*Cossart et al., 2003*]). (**E**) MEC LII Pyramidal-Multiform cell with axons in layers II–VI and dendrites in layers I-III (NMO_07252; (*Quilichini et al., 2010*)). (**F**) Axon and dendrite locations for 100 representative neuron types (glutamatergic: black; GABAergic: gray; full matrix: hippocampome.org/morphology or hippocampome. org/php/images/morphology/Morphology_Matrix.jpg).

given different names due to varying emphases of researchers. For instance, CA1 Radiatum cells (*Cope et al., 2002*) have also been called CaBP-positive non-pyramidal cells (*Toth and Freund, 1992*) and Schaffer collateral associated cells (*Vida, 2010*). The authors' original names are always included as synonyms of the Hippocampome.org-assigned identifiers (see 'Materials and methods') to facilitate tracing information provenance and for ease of communication.

An advantage of defining neuron types by their axonal and dendritic patterns is the practical cross-species invariance of these features, at least within rodents. Hippocampome.org collates data from all healthy young or adult rats, mice, and guinea pigs, but all evidence is meta-annotated with species and age information to allow separate tracking. The few cross-species morphological differences identified so far, such as the extent of CA3c infrapyramidal mossy fibers (*Blaabjerg and Zimmer, 2007*), are immaterial to the current granularity of the knowledge base.

## Major neurotransmitter distinctions

Hippocampal neurons mainly release glutamate or gamma-aminobutyric acid (GABA) (*Kullmann, 2007*). When the literature does not present explicit evidence for determining the neurotransmitter, a neuron type is deemed putatively glutamatergic or GABAergic depending on ancillary characteristics including (a)symmetry of synapses, excitatory or inhibitory effect, relative somatic abundance within an area, presence of dendritic spines, and local or projecting nature of the axons. Although exceptions exist for all these criteria, taken together, they are sufficiently indicative to enable the tentative inference of the main neurotransmitter for all identified neuron types based on published information (*Figure 1F*: black, presumed glutamatergic; gray, presumed GABAergic).

Neurons with the same axonal and dendritic patterns, but different neurotransmitters, belong to different types. Interestingly, only one such case is known: in CA1, Cajal–Retzius cells, which were recently characterized as glutamatergic and more abundant than previously assumed in adult rats (*Quattrocolo and Maccaferri, 2014*), and (GABAergic) Neurogliaform cells (*Price et al., 2005*) have both axons and dendrites confined to SLM, though with extremely different arbor densities and shapes. Thus, putatively excitatory and inhibitory neurons in the hippocampus tend to have completely distinct morphologies.

Although most excitatory neurons in all hippocampal areas have long-range axons, Hippocampome.org also includes several local glutamatergic types in DG, CA3, and CA1. Similarly, a substantial minority of GABAergic types in these areas (16/71) have projecting axons. Knowledge is sparser in CA2, Sub, and EC. The axons of several presumed glutamatergic types in EC have not been reconstructed beyond layer VI or the angular bundle, so their participation in long-range pathways remains largely unknown (*Canto and Witter, 2012a*, *2012b*).

## Synaptic specificity

Certain interneurons selectively discriminate between GABAergic and glutamatergic post-synaptic partners. Like the neurotransmitter, preferential connectivity profoundly affects circuit function. This characteristic is thus essential to distinguish neuron types with the same axonal–dendritic pattern. For example, Axo-axonic and Basket cells in CA3 and CA1 have dendrites spanning all layers and axons limited to SP (e.g., *Freund and Buzsáki, 1996*; *Klausberger and Somogyi, 2008*). Axo-axonic cells, however, contact exclusively Pyramidal cells on the axon initial segments (*Li et al., 1992*; *Ganter et al., 2004*), whereas Basket cells synapse perisomatically on both principal neurons and interneurons (*Buhl et al., 1994*; *Vida, 2010*). The separation of Axo-axonic and Basket cells yields two additional types with the same axonal–dendritic patterns.

Several 'interneuron-specific' neuron types preferentially target other GABAergic neurons over glutamatergic types. Since all known cases correspond to unique morphological patterns, such selectivity does not separate further neuron types. However, this core property is indicated in their adopted names, as in CA1 Interneuron-specific O-R cells (cf. *Gulyás et al., 1996*).

## Other properties distinguishing neuron types

The hippocampal literature offers a wealth of biochemical and electrophysiological data (see below) that are linked in the knowledge base to specific morphologically defined neuron types ('Materials and methods'). Differences in *individual* molecular or biophysical features, such as the presence/absence of a single biomarker or the high/low value in one membrane property, are frequently reported for a given axonal–dendritic pattern. These cases are consistently annotated in Hippocampome.org as indicating potential sub-types, but do not establish full neuron types.

In contrast, when opposite expression of multiple markers and large discrepancies in several membrane properties strongly support the existence of distinct neuron types by converging molecular and electrophysiological evidence, the same morphological pattern is divided into two types. For example, one type of CA1 Basket cells expresses parvalbumin (PV) and μ-opioid receptor, but not cholecystokinin (CCK), cannabinoid receptor 1 (CB1), and substance P receptor (sub P rec), while another type displays the opposite expression profile (*Pawelzik et al., 2002*); moreover, PV+ cells have significantly lower input resistance ($R_{in}$), faster membrane time constant ($\tau_m$), narrower action potential width ($AP_{width}$), and smaller slow after-hyperpolarization (sAHP) than CCK+ cells (*Bartos and Elgueta, 2012*).

Similarly, both CA1 Bistratified and Ivy cells have axons and dendrites extending from SO to SR, but the former are positive for PV and somatostatin (SOM) and negative for neuronal nitric oxide synthase (nNOS), while the opposite holds for the latter; furthermore, relative to Ivy cells, Bistratified cells have lower firing threshold ($V_{thresh}$) and maximum firing rates (maxFR), larger action potential amplitude ($AP_{ampl}$), and narrower $AP_{width}$ (*Fuentealba et al., 2008*; *Tricoire et al., 2011*). Equally convincing data support the distinction of CA3 Ivy and Bistratified cells as well as the separation of basket cells in DG and CA3. In all, five additional types with non-unique axonal–dendritic patterns are established on the basis of clear and substantial molecular and electrophysiological information.

## Molecular markers

Expression data are obtained from studies detecting proteins (immunohistochemistry) or mRNA (in situ hybridization, promoter constructs, or single-cell RT-PCR). The knowledge base has information on 96 biomarkers, and positive or negative expression of at least one biomarker is known for two-thirds (81/122) of the neuron types (*Figure 2*; extended listing: hippocampome.org/markers). The most cited biomarker is parvalbumin (PV), appearing in ~25% of the biomarker sources in Hippocampome.org, and whose expression is known for 56 neuron types.

In 79 of the 566 cases with available expression data for the 20 most cited biomarkers (non-gray squares in *Figure 2* and hippocampome.org/markers), evidence exists for a neuron type to both express and not express a given biomarker. In 13 of these cases, the incongruity can be attributed to species, technique, or sub-cellular localization differences (*Figure 2*, orange, down-facing flags). For example, mossy cells express calretinin (CR) in mice (*Blasco-Ibáñez and Freund, 1997*), but not in rats (*Freund et al., 1997*). In another case, in situ hybridization showed $GABA_A$-α1 expression in rat CA1 Pyramidal cells (*Miralles et al., 1994*), yet somatic immunohistochemistry of these neurons is negative; however, their dendrites show positive expression (*Lopez-Tellez et al., 2004*).

Another 58 mixed cases are designated as possible subtypes (*Figure 2*, blue and green flags in the same square), when in a population of cells with equivalent morphology, from the same experiment, a sizeable proportion expresses a biomarker and a sizeable proportion does not. The interpretation of potential subtypes is sometimes supported by distinct somatic locations. For example, CA3 Pyramidal cells are positive for chicken ovalbumin upstream promoter transcription factor II (COUP-TFII) in the temporal, but not septal hippocampus (*Fuentealba et al., 2010*). Similarly, superficial CA1 Pyramidal cells express calbindin (CB) while those deep in CA1 SP do not (*Jinno et al., 1999*; *Sadowski et al., 2002*).

In the remaining eight cases of mixed expression (red, upward-facing flags), data come from multiple sources that use the same species and technique (e.g., rat immunohistochemistry); however, non-identical experimental details (e.g., the antibodies used) prevent conclusive interpretation.

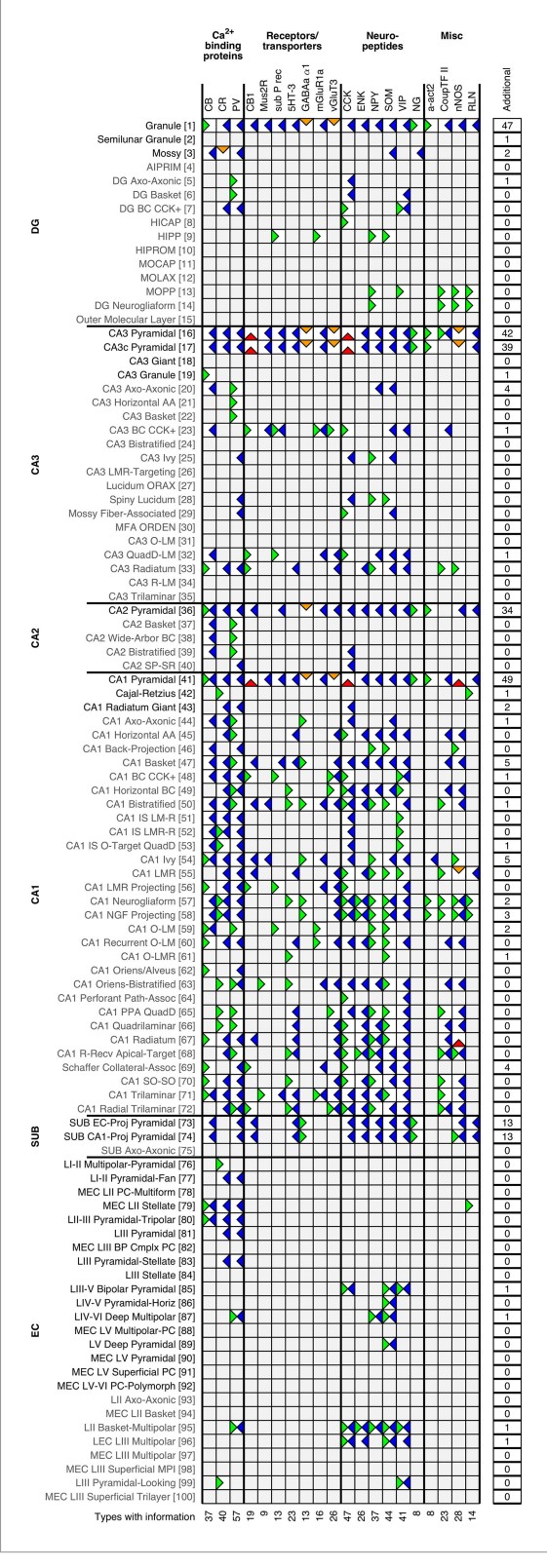

**Figure 2**. Expression of 20 common biomarkers for 100 representative neuron types (full matrix: hippocampome. org/markers or hippocampome.org/php/images/marker/Marker_Matrix.jpg; complete list of abbreviations: hippocampome.org/help). Positive expression: left green flags; negative expression: right blue; mixed expression (possible subtypes): left/right green blue; mixed expression (different experimental protocols, species, or

*Figure 2. continued on next page*

*Figure 2. Continued*

sub-cellular localization): top orange; unresolved mixed expression: bottom red; empty gray boxes indicate that morphologically linkable information was not found. The right summary column reports the number of additional biomarkers with known expression for each neuron type. Bottom values are counts of neuron types with available information for each biomarker.

## Electrophysiology

Electrophysiological characteristics vary extensively across hippocampal neurons. Passive, spike, and other response parameters are commonly recorded in the slice preparation. Slice recording is highly dependent on experimental method (patch clamp or sharp microelectrode), animal species and age, bathing and intra-electrode solution, and temperature. Thus, summary values (*Table 1* for 50 representative neuron types; complete listing: hippocampome.org/electrophysiology) are not averaged across all studies, but rather reflect data from a single report under 'preferred' conditions (operationally defined as rat patch clamp at body temperature) with the largest number of measurements (n). These settings represent 61% of the data in *Table 1*. If these conditions are unavailable, the precedence order is patch clamp over microelectrodes, body temperature over room temperature, and rats over mice over guinea pigs. However, *all* collected parameter values (preferred or otherwise) for every neuron type are accessible at Hippocampome.org, each complete with metadata, available statistics, and direct links to underlying experimental evidence. Three-quarters of the neuron types (93/122) have measurements reported for four or more parameters, the most common being $AP_{ampl}$, fast afterhyperpolarization amplitude (fAHP), and $V_{thresh}$.

## Summary of design criteria

Although Hippocampome.org primarily categorizes neuron types according to their morphologies (locations of axons and dendrites across the 26 parcels of the hippocampal formation), several other properties are also considered (*Figure 3*). When evaluating publications describing a neuron type, we first apply a set of interpretive rules to the morphological information (see 'Materials and methods' and hippocampome.org/full-interp). We then follow a series of systematic criteria that can result in one of three end points: the mined information is consistent with (and supplementary of) a currently active, fully defined neuron type (hippocampome.org/supplemental); the information is distinct enough to establish a new active neuron type; or the information is insufficient, thus generating an 'on hold' neuron type (hippocampome.org/on-hold).

## A digital storehouse of explicit knowledge

Hippocampome.org provides free, user-friendly online access to the entire information content. The annotated evidence supports 3697 distinct 'pieces of knowledge' (PoK) regarding the presence or absence of axons or dendrites within any of the 26 parcels (e.g., *Granule cells axons are found in CA3 SL* represents one PoK), the expression or non-expression of a biomarker (e.g.,*Granule cells are positive for CB*), and individual electrophysiological properties (e.g., *the $AP_{width}$ of Granule cells is 1.71 ± 0.58 ms, mean ± standard deviation*). This knowledge exceeds by 2–3 orders of magnitude (*Figure 4A*); the average numbers of PoK collated from relevant articles (~5) and reviews (~27). Integration of published information from multiple sources, by linking together molecular, electrophysiological, and morphological data, produces an explicit knowledge web of references, properties, and neuron types interlaced in a many-to-many fashion (*Figure 4B*). Of equal importance, this resource explicitly highlights the non-uniform distribution of knowledge among neuron types and across morphological, molecular, and electrophysiological properties (*Figure 4C*). The approximate ratio of morphological, molecular, and electrophysiological PoK is 5:3:2. While CA1 is the most studied hippocampal area (1321 PoK), outstanding knowledge gaps remain on interneuron diversity in subiculum (only Axo-axonic cells have enough information for inclusion as an active neuron type in Hippocampome.org) and EC (no GABAergic cells characterized with either soma or axons in deep layers).

The massive amount of knowledge and online availability of Hippocampome.org enable both broad-scope analytics and quick-use information checks. All information can be browsed or searched.

Table 1. Electrophysiological properties for 50 representative neuron types (full table: hippocampome.org/electrophysiology or hippocampome.org/php/images/electrophysiology/Electrophysiology_Table.jpg)

| | | $V_{rest}$ (mV) | $R_{in}$ (MΩ) | $\tau_m$ (ms) | $V_{thresh}$ (mV) | fAHP (mV) | $AP_{ampl}$ (mV) | $AP_{width}$ (ms) | maxFR (Hz) | sAHP (mV) | Sag ratio |
|---|---|---|---|---|---|---|---|---|---|---|---|
| DG (8) | Granule | −75 ± 2(16) ∈ 5 | 228 ± 79.1 (31) ∈ 5 | 26.9 ± 6.7 (31) ∈ 5 | 35 ∈ 5 | 11.7 ± 1.1 (16) ∈ 3 | 91 ∈ 5 | 0.87 ± 0.06 (16) ∈ 2 | 72 ± 8(16) | 0.6 ± 0.4 (16) ∈ 3 | 0.97 ± 0.01 (16) ∈ 3 |
| | Mossy | −62 ± 1(9) | 199 ± 19(9) | 41 ± 3(8) | 23.7 | 6.2 ± 0.9(9) | 62.5 | 0.78 ± 0.04(9) | 50 ± 6(9) | 2.8 ± 0.7(9) | 0.81 ± 0.03(9) |
| | AIPRIM | −64 ± 2(4) | 363 ± 62(4) | 30 ± 6(4) | 16 | 12.3 ± 2.3(4) | [76, 77.9] | 0.5 ± 0.02(4) | 81 ± 9(4) | 9 ± 2.7(4) | 0.8 ± 0.04(4) |
| | DG Axo-axonic | −65.1 ± 3.9(14)e | 73.9 ± 23.8(14)e | 7.7 ± 3.8(14)e | 13r ∈ 2 | 7r ∈ 2 | 78r ∈ 2 | 0.42r ∈ 2 | 85r | 4.5r ∈ 2 | – |
| | DG Basket | −62 ± 3(3) | 43 ± 5(3) ∈ 3 | 10 ± 1(3) | [17.6, 19.0] ∈ 3 | 20 ± 2.3(3) ∈ 2 | [71.6, 73.6] ∈ 2 | 0.25 ± 0.04 (3) ∈ 2 | 230 ± 15 (3) ∈ 2 | 2.3 ± 0.2(3) | 0.97 ± 0.02 (3) ∈ 2 |
| | HIPROM | −65 ± 6(3) | 371 ± 47(3) | 35 | [25.1, 27.3] | 13.1 ± 3.0(3) | 80.8 | 0.72 ± 0.08(3) | 69 ± 4(3) | 3.1 ± 1.0(3) | 0.82 ± 0.02(3) |
| | MOLAX | −54.5 ± 1.9 (13) ∈ 2 | 198.2 ± 23.8 (13) ∈ 2 | 18.4 ± 1.1(13) | 15.2 ∈ 2 | 11 ∈ 2 | [43.2, 44.4] ∈ 2 | [1.26, 1.64] | 50 | 3.5mr ∈ 2 | – |
| | Total Molecular Layer | −54.5 ± 1.9(13) | 198.2 ± 23.8(13) | 18.4 ± 1.1(13) | 15.2 | 11 | [43.2, 44.4] | [1.26, 1.64] | 50 | 4.5 | – |
| CA3 (8) | CA3 Pyramidal | −60.5 ± 5.4 (43) ∈ 8 | 126 ± 8(35) ∈ 8 | 61 ± 24(36) ∈ 7 | 13 ∈ 5 | 10.2 ± 0.5 (43) ∈ 3 | 97.6 ± 1.9 (43) ∈ 4 | 1 ± 0.1(43) ∈ 4 | 40 ± 20.8(3) | 7.5m ∈ 3 | 1.01 ± 0.01 (7) ∈ 4 |
| | CA3 Giant | −57 ± 1.2(28)r | 595 ± 224(28)r | 67 ± 23(28)r | 22r | 14r | 76 ± 7.5(28)r | 1.1 ± 0.1(28)r | >50 ± 3(13) | 7.1r | 0.68r |
| | CA3 Granule | −78 ± 0.5(15) | 139 ± 11(15) | 17.1 ± 1.8(15) | 27.4 | 7.95 | 104.3 | [0.821, 1.072] | >100 | 1.75 | 0.99 |
| | CA3 Basket | −58.5 ± 2.8 (6)mr | 122.9 ± 26.7 (6)mr | 11.2 ± 2.9 (6)mr | 25.6mr | 35mr | 77.1mr | 0.54 ± 0.1 (8)mr | 33.6 ± 6.4 (6)mr | 3mr | 0.93mr |
| | Lucidum ORAX | −61 ± 6(9) ∈ 2 | 284 ± 180(9) ∈ 2 | 42 ± 17(8) ∈ 2 | 16.3 ∈ 2 | 15 ± 6.6(9) ∈ 2 | 77.7 ∈ 2 | 0.53 ± 0.2 (9) ∈ 2 | 75 ± 28.3 (8) ∈ 2 | 6.6 ± 3.9 (9) ∈ 2 | 0.9 ± 0.1(8) ∈ 2 |
| | MFA ORDEN | −57 ± 5(13) | 225 ± 93(13) | 29.1 ± 14.6(13) | 20 | 13.1 ± 1.9(13) | 74 | 0.72 ± 0.15(13) | 73 ± 16(13) | 6.5 | – |
| | CA3 O-LM | −60 ± 12 (15)mr | 315.1 ± 161.1(15)mr | 33.3 ± 5.4 (15)mr | [16, 37]mr | 34.8mr | 109mr | 0.84 ± 0.2 (15)mr | 182 ± 161.1 (15)mr | 1.3mr | 0.79 ± 0.1 (10)mr |
| | CA3 Trilaminar | −61.2 ± 13 (8)mr | 167.3 ± 59.1 (8)mr | 16.9 ± 8.8 (8)mr | 12mr | 30mr | [69, 99]mr | 0.57 ± 0 (8)mr | 101.5 ± 62.2(8)mr | 0.1mr | 0.88mr |
| CA2 (4) | CA2 Basket | −71.2 ± 4(6)e | 77 ± 19.3(6)e | 8.2 ± 3.2(6)e | [17, 27]e | 22 ± 5.1(6)e | 62.7 ± 9(6)e | 0.5 ± 0.1(6)e | >180e | 3e | 0.99e |
| | CA2 Wide-arbor BC | −74.9 ± 5.6(10)e | 111.8 ± 36.7(10)e | 12.6 ± 4.2(10)e | [26, 34]e | 19.5 ± 9(10)e | 65.5 ± 7.1(10)e | 0.6 ± 0.1(10)e | >125e | 7e | 0.55e |
| | CA2 Bistratified | −72.7 ± 1.1(3)e | 83.3 ± 16.7(3)e | 13.7 ± 10(3)e | [42, 56]e | 14.4 ± 11.8(3)e | 65.2 ± 7.8(3)e | 0.4 ± 0.1(3)e | – | 3.2e | 0.99e |
| | CA2 SP-SR | −71 ± 4.5(8)e | 82.6 ± 24.4(8)e | 12.7 ± 3.8(8)e | 19e | 11.5e | 67e | 0.5 ± 0.1(8)e | >160e | 3.3e | 0.86e |
| CA1 (16) | CA1 Pyramidal | −62.4 ± 2.4 (21) ∈ 7 | 65.6 ± 4.4 (20) ∈ 7 | 22.4 ± 1.5 (20) ∈ 6 | 16.6 ∈ 7 | 6.8 ∈ 2 | 90 ∈ 4 | 1 ∈ 3 | >32 | 1.7 ∈ 3 | 0.74 ∈ 3 |
| | CA1 Radiatum Giant | −66 ± 3.6 (16) ∈ 2 | 56 ± 14(25) ∈ 2 | 50 ± 7.9(7)r ∈ 2 | 31 ∈ 2 | 8 ∈ 2 | 70 ∈ 2 | 1.9 ± 0.2(7)r | >26 | 8.3r ∈ 2 | 0.86 ∈ 2 |
| | CA1 Horizontal AA | −57 ± 5 (15)m | [186, 252]r | [16, 32]r | 25m | [6.0, 13.7]r | [71.4, 90.2]r | [0.7, 1.1]r | 87r ∈ 2 | 6.1m | 0.74m |
| | CA1 Basket | −57 ± 5(15) m ∈ 2 | 116 ± 63(15) m ∈ 2 | 13 ± 8(15)m ∈ 2 | 19 ∈ 3 | 10 ∈ 3 | 62 ∈ 3 | 0.54 ± 0.11 (15)m ∈ 2 | >60 | 2e ∈ 3 | 0.84 ± 0.06 (15)m |
| | CA1 BC CCK+ | −61.4 ± 3.2(5) | 281.68 ± 79.7(5) | 25.07 ± 5.6(5) | 21.5 ∈ 3 | 15.17 ± 3.4 (5) ∈ 3 | 76.92 ± 11.7(5) ∈ 3 | 0.84 ± 0.1(5) | >30 | 7 ∈ 2 | 0.825 |

*Table 1. Continued on next page*

Table 1. Continued

| | | V$_{rest}$ (mV) | R$_{in}$ (MΩ) | τ$_m$ (ms) | V$_{thresh}$ (mV) | fAHP (mV) | AP$_{ampl}$ (mV) | AP$_{width}$ (ms) | maxFR (Hz) | sAHP (mV) | Sag ratio |
|---|---|---|---|---|---|---|---|---|---|---|---|
| | CA1 Horizontal BC | −55.4 ± 9.5 (17) ∈ 2 | [116, 199] ∈ 2 | [15.4, 25.5] ∈ 2 | 24 ∈ 2 | 11 ∈ 2 | 130 ∈ 2 | 0.77 ± 0.10 (18)m ∈ 2 | >50m | 4.8m ∈ 2 | 0.6 ∈ 2 |
| | CA1 Ivy | −71 | 72.8 ± 53.6(5) | 7.6 ± 4.1(5) | 30.1 | 13.6 ± 3.8(5) | 44 | 0.8 ± 0.2(5) | – | 3 | 0.98 |
| | CA1 LMR | −53.1 ± 4.0 (48)$_r$ ∈ 3 | 352 ± 107 (49)$_r$ ∈ 3 | 32.9 ± 12.7 (11)$_r$ ∈ 3 | 13.2$_r$ ∈ 3 | 21.5$_r$ ∈ 3 | 86.9 ± 11.0 (49)$_r$ ∈ 3 | 1.3$_r$ ∈ 3 | – | 0.2$_r$ ∈ 4 | 0.92 ± 0.11 (15)$_r$ ∈ 3 |
| | CA1 Neurogliaform | −63.1 ± 5.6 (33) ∈ 2 | 215.3 ± 92.8 (32) ∈ 2 | 12.43 ± 4.59(32) ∈ 2 | 32.4 ∈ 2 | 20.4 ± 4.1 (34) ∈ 2 | 73 ∈ 2 | 0.9 ± 0.18 (26) ∈ 2 | 52.8 ± 31.0(26) | 9 ∈ 2 | 0.99 ∈ 2 |
| | CA1 NGF Projecting | −63.1 ± 5.6(33) | 215.3 ± 92.8(32) | 12.43 ± 4.59(32) | 32.4 | 20.4 ± 4.1(34) | 73 | 0.9 ± 0.18(26) | 52.8 ± 31.0(26) | 9 | 0.99 |
| | CA1 Recurrent O-LM | [−85, −65]$^e$ | 70 ± 13.72(8)$^e$ | 12.8 ± 1.5(8)$^e$ | 25$^e$ | 16.1 ± 10.7(8)$^e$ | 58.75 ± 7.2(8)$^e$ | 0.6 ± 0.3(8)$^e$ | >150$^e$ | 5$^e$ | 0.71$^e$ |
| | CA1 PPA QuadD | −64 ± 7 (23)m | 216 ± 124 (23)m | 46 ± 18 (23)m | 27m | 22 ± 3(23)m | 61 ± 7(23)m | 0.77 ± 0.12 (23)m | >40m | 1.3m | 0.79 ± 0.09 (23)m |
| | Schaffer Collateral-Assoc | −55.8 ± 2.8 (10)$^e$ ∈ 2 | 96.3 ± 36.0 (10)$^e$ ∈ 2 | 16.2 ± 8.9 (10)$^e$ ∈ 2 | 11$^e$ ∈ 2 | 11.4 ± 2.8 (10)$^e$ ∈ 2 | 70.8 ± 8.0 (10)$^e$ ∈ 2 | 0.74 ± 0.1 (10)$^e$ ∈ 2 | >100 | 5.5$^e$ ∈ 2 | 0.86$^e$ |
| | CA1 SO–SO | −59 ± 10 (19)m ∈ 2 | 401 ± 212 (19)m | 38 ± 13 (19)m | 24m ∈ 2 | 15 ± 4(19)m ∈ 2 | 48 ± 10(19) m ∈ 2 | 1.12 ± 0.14 (19)m | >160m | 4.8m ∈ 2 | 0.69m ∈ 2 |
| | CA1 Trilaminar | −64 ± 7 (23)m | 216 ± 124 (23)m | 46 ± 18 (23)m | 27m | 22 ± 3(23)m | 61 ± 7(23)m | 0.77 ± 0.12 (23)m | >130m | 4.7m | 0.79m |
| | CA1 Radial Trilaminar | −57 ± 5 (15)m | 116 ± 63 (15)m | 13 ± 8(15)m | 29m ∈ 3 | 25 ± 4(15)m ∈ 3 | 48 ± 8(15)m ∈ 3 | 0.54 ± 0.11 (15)m | >120m | 0.82m ∈ 3 | 0.89m |
| EC (14) | LI-II Multipolar-Pyramidal | [−70, −56] | 430 ± 121.7(37) | 25 ± 15.2(37) | [8.1, 23.1] | [3.7, 7.3] | 77.5 ± 15.2(37) | 1.22 ± 0.4(37) | >50 | 4.7 | 0.78 ± 0.06(37) |
| | LI-II Pyramidal-Fan | [−62, −59] ∈ 2 | 400 ± 98(96) ∈ 2 | 15.8 ± 14.2 (96) ∈ 2 | [7.5, 8.4] ∈ 2 | [7.1, 7.3] ∈ 2 | 69 ± 19.6 (96) ∈ 2 | 1.25 ± 0.49 (96) ∈ 2 | >40 | [0, 6] ∈ 2 | 0.66 ± 0.05 (96) ∈ 2 |
| | MEC LII PC-Multiform | [−70, −56] | 430 ± 121.7(37) | 25 ± 15.2(37) | [8.1, 23.1] | [3.7, 7.3] | 77.5 ± 15.2(37) | 1.22 ± 0.4(37) | >25 | 4.7 | 0.78 ± 0.06(37) |
| | MEC LII Oblique Pyramidal | [−62, −59] | 400 ± 98(96) | 15.8 ± 14.2(96) | [7.5, 8.4] | [7.1, 7.3] | 69 ± 19.6(96) | 1.25 ± 0.49(96) | >40 | [0, 6] | 0.66 ± 0.05(96) |
| | MEC LII Stellate | [−62, −59] ∈ 2 | 30.2 ± 12.5 (112)$^e$ ∈ 2 | 8.9 ± 1.9 (112)$^e$ ∈ 2 | [7.5, 8.4] ∈ 2 | [7.1, 7.3] ∈ 2 | 69 ± 19.6 (96) ∈ 2 | 1.25 ± 0.49 (96) ∈ 2 | >40 | [0, 6] | 0.66 ± 0.05 (96) ∈ 2 |
| | LII-III Pyramidal-Tripolar | [−70, −56] ∈ 2 | 400 ± 98(96) ∈ 2 | 15.8 ± 14.2 (96) ∈ 2 | [8.1, 23.1] ∈ 2 | [3.7, 7.3] ∈ 2 | 69 ± 19.6 (96) ∈ 2 | 1.25 ± 0.49 (96) ∈ 2 | >40 | 4.7 ∈ 2 | 0.66 ± 0.05 (96) ∈ 2 |
| | LEC LIII Multipolar Principal | [−68, −65] | 450 ± 78(27) | 29 ± 12.5(27) | [21.17, 26.03] | [10.23, 11.04] | 70 ± 15.6(27) | 1.38 ± 0.26(27) | >25 | 4.06 | 0.9 ± 0.1(27) |
| | LEC LIII Complex Pyramidal | [−68, −65] | 450 ± 78(27) | 29 ± 12.5(27) | [21.17,26.03] | [10.23,11.04] | 70 ± 15.6(27) | 1.38 ± 0.26(27) | >25 | 4.06 | 0.9 ± 0.1(27) |
| | LIII Pyramidal-Stellate | [−68, −65] ∈ 2 | 450 ± 78(27) ∈ 2 | 29 ± 12.5 (27) ∈ 2 | [21.17, 26.03] ∈ 2 | [10.23, 11.04] ∈ 2 | 70 ± 15.6 (27) ∈ 2 | 1.38 ± 0.26 (27) ∈ 2 | [85, 105]$^e$ ∈ 2 | 4.06 ∈ 2 | 0.9 ± 0.1(27) |
| | LIII-V Bipolar Pyramidal | [−69, −67] | 490 ± 79(28) | 36 ± 16(28) | [18.2, 21.8] | [5.6, 8.7] | 70 ± 15.9(28) | 1.48 ± 0.2(28) | >29 | [2, 7] | 0.86 ± 0.11(28) |
| | LIV-VI Deep Multipolar | −59.8 ± 6.8 (8) ∈ 4 | 272.3 ± 105.8(8) ∈ 4 | 26.6 ± 6.5 (8) ∈ 4 | 36 ∈ 4 | 15 ∈ 4 | 75 ∈ 4 | 1.39 ± 0.6 (8) ∈ 4 | >70 | 4.4 ∈ 4 | 0.9 ∈ 3 |
| | LV Deep Pyramidal | −65.02 ± 3.8(38)$^e$ ∈ 2 | 75.67 ± 28.72(38)$^e$ ∈ 2 | 12.83 ± 3.65(38)$^e$ ∈ 2 | 22.14$^e$ ∈ 2 | 17.44 ± 2.64 (38)$^e$ ∈ 2 | 64.96 ± 6.53(38)$^e$ ∈ 2 | 1.53 ± 0.28 (38)$^e$ ∈ 2 | >70 | 0.1$^e$ ∈ 2 | 0.88$^e$ ∈ 2 |
| | MEC LV-VI PC-Polymorph | −63 | 480 ± 94(14) | 41 ± 19(14) | [26.7, 28.7] | [7.1, 9.1] | 73 ± 8 (14) | 2.1 ± 0.8(14) | >33 | 1.6 | 0.91 ± 0.09(14) |
| | LEC LVI Multipolar-PC | −64 | 450 ± 90(13) | 26 ± 14(13) | [26.8, 29.0] | [6.0, 8.8] | 74 ± 14 (13) | 1.7 ± 0.4(13) | >10 | 1.2 | 0.9 ± 0.1(13) |

Values are mean values ± standard deviations or [range]; parentheses indicate the number of data points (default = 1; 2 for ranges). Values of maxFR reported as relative lower limits (e.g., >30 Hz) are measured from limited spike trains. The data are selected from a set (∈) of available values based on experimental conditions and number of measurements. Preferred conditions (rat/body temperature/patch clamp) are not indicated; otherwise, m = mice, r = room temperature, e = microelectrodes.

Browsing starts with summaries of morphological (hippocampome.org/morphology), molecular (hippocampome.org/markers), or electrophysiological (hippocampome.org/electrophysiology) data, similar to hyperlinked versions of *Figures 1F, 2*, and *Table 1*, respectively. Searching for neuron types is accomplished through selections from dynamically updated pull-down menus of any combination of properties, such as 'axons located in DG', 'PV-negative', and 'V$_{thresh}$ > 20 mV' (*Figure 5A*). Compound queries with AND & OR Boolean connectors can uncover unexpected results even for the most experienced hippocampal researchers. Searches can also be conducted by PubMed identifiers or author names (*Figure 5B*). Returned results contain full bibliographic information, a link to the article, and a list of neuron types with information from the article.

Both browsing and searching lead to summaries of all information associated with a given neuron type (*Figure 5C*): synonyms, morphology, electrophysiology, biomarkers, a representative figure, and known pre- and post-synaptic connectivity (see below). Every property on each neuron page or browse summary links to an evidence page that lists all supporting bibliographic citations complete with extracted quotes, figures, tables, and exact pointers to the relevant pages and paragraphs

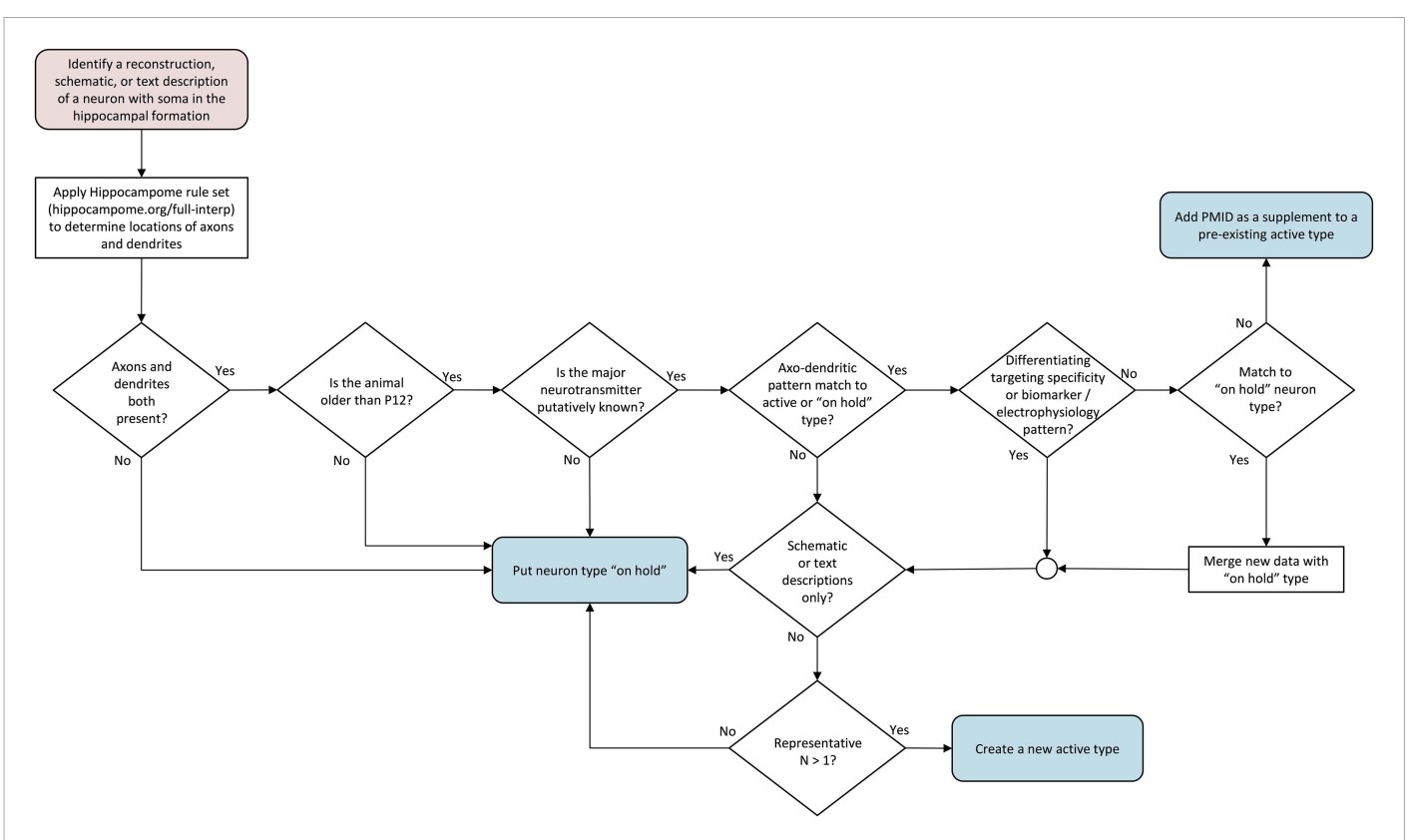

**Figure 3**. Flow chart of inclusion criteria for neuron types. Beginning with a reconstruction, schematic, or text description of a neuron morphology, the flow chart ends with either a new 'on hold' neuron type, supplemental information for an existing active neuron type, or a new active neuron type. Intermediate decision points evaluate the presence of both axons and dendrites, the determination of the main neurotransmitter, the uniqueness of the new type, and whether information is sufficient to create a new active type.

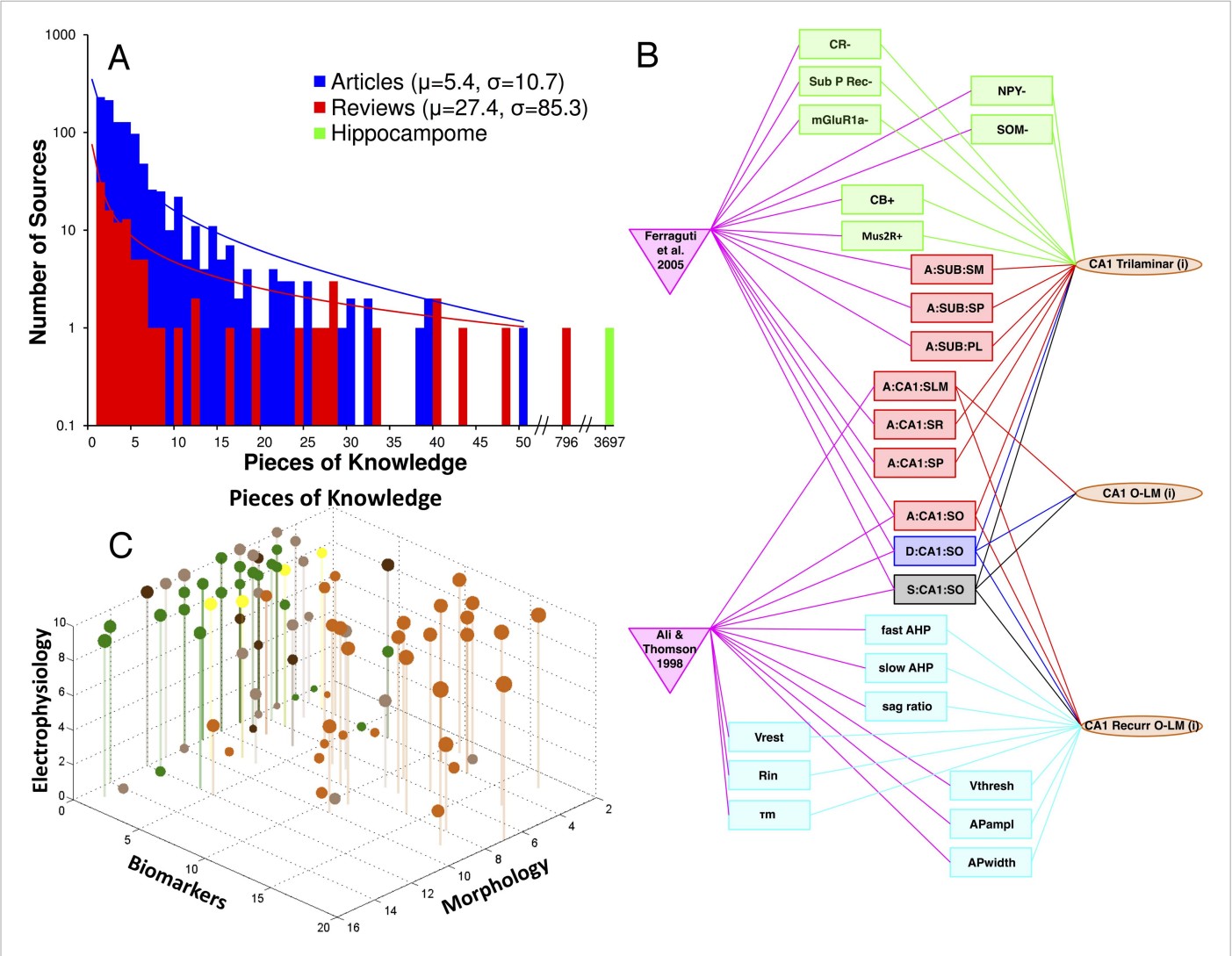

**Figure 4**. Quantifying knowledge in Hippocampome.org about morphology, biomarkers, and electrophysiology of hippocampal neuron types. (**A**) Histograms comparing the sum of pieces of knowledge (PoK) in relevant journal articles or book chapters, in reviews, and in Hippocampome.org. (**B**) Interconnected knowledge graph of neuron type properties mined from two typical journal articles. (**C**) Balloon plot of collated knowledge for a majority of GABAergic neuron types. The balloon size indicates the sum of PoK for that type across all three dimensions; balloon color denotes the subregion (as in *Figure 1*). Note the dearth of biomarker information in entorhinal cortex (EC) and the uneven distribution of data between CA3 and CA1.

(*Figure 5D*). Hippocampome.org contains 13,888 pieces of evidence, including all known sources for many neuron types and properties, but only an adequate number of sources to firmly support established knowledge (e.g., we have only annotated a fraction of all published evidence that granule cells extend axons in CA3 SL).

## Pairwise correlations

Knowledge integration facilitates the discovery of relations between neuronal properties that would ordinarily remain hidden in the scattered literature. Most PoK in Hippocampome.org, such as axonal presence in a layer or expression of a biomarker, are categorical, and their statistical co-occurrence can be analyzed with contingency tables ('Materials and methods'). To allow comparison across experiments, electrophysiological parameters are converted (for this analysis only) from continuous to categorical variables by labeling values in the top and bottom one-third of the range, respectively, as high and low. This approach reveals several interesting relationships (*Box 1*): for example, positive or

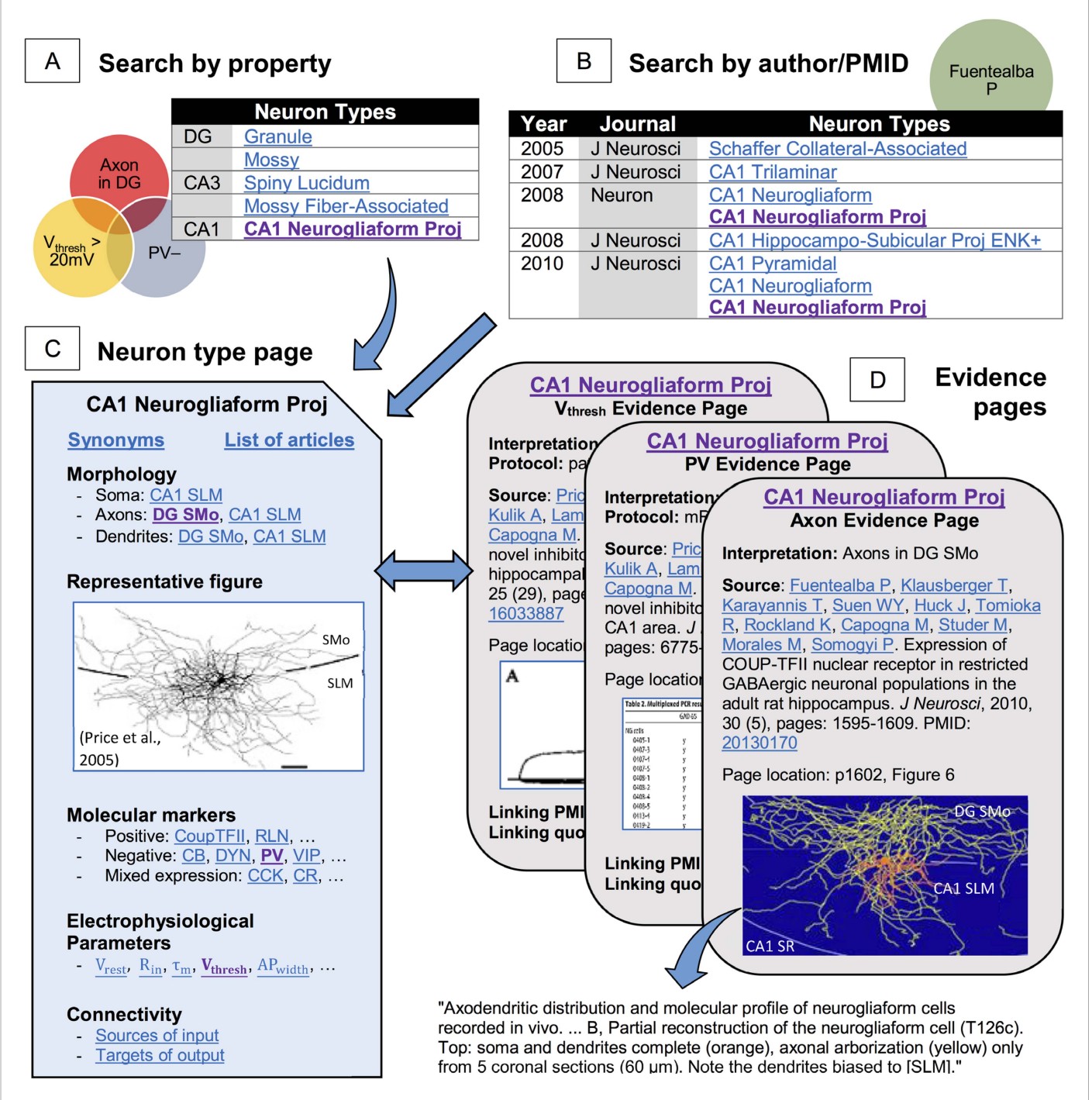

Figure 5. Hippocampome.org enables searching neuron types by neurotransmitter; axon, dendrite, and soma locations; molecular expression; electrophysiological parameters; and input/output connectivity. (A) Sample query for parvalbumin-negative neuron types with axons in DG and firing threshold >20 mV. (B) The knowledge base may also be queried for a specific PubMed ID or author name (e.g., 'Fuentealba P'). (C, D) Returned results link to (C) neuron type summary pages (*Figure 1A* from *Price et al., 2005*, *J. Neurosci.* 25:6775–6786 [permission to reuse granted by SfN]) and (D) evidence from published figures, tables, and text quotes supporting all reported properties (Figure 6B from *Fuentealba et al., 2010*, *J. Neurosci.* 30:1595–1609; Table 2 and Figure 3A from *Price et al., 2005*, *J. Neurosci.* 25:6775–6786 [permission to reuse granted by SfN]).

negative expression of neuropeptide Y (NPY) tends to co-occur, respectively, with high or low fAHP (p < 0.001 with Barnard's exact test; post hoc t-test p < 0.02 with real values from all n = 42 pieces of evidence); vasoactive intestinal polypeptide (VIP) is mutually exclusive with CB (p < 0.04); and of

## Box 1. Trends assessed from data collated in the Hippocampome.

The trend across clear expression patterns of calbindin (CB), calretinin (CR), and parvalbumin (PV) supports the general idea that only one of these calcium binding proteins is expressed at a time within a cell. While CB is never co-expressed with either CR ($p < 0.05$, $n = 19$) or PV (NS, $n = 31$), PV and CR do co-occur in 2 out of 12 neuron types known to express at least one of them ($p > 0.49$, $n = 31$). A less known but more stringent relationship exists between CB and vasoactive intestinal peptide (VIP): among the 24 neuron types for which the expression information on both of these markers is known, 14 are positive for either CB or VIP, but none express both ($p < 0.04$, $n = 24$).

Clearly positive expression of neuropeptide Y (NPY) tends to coincide with very high values of the fast afterhyperpolarization (fAHP) ($p < 0.001$, $n = 23$) and membrane firing threshold ($V_{thresh}$) ($p < 0.005$, $n = 21$), and clearly negative expression tends to be associated with very low values of both parameters. fAHP is similarly related also to expression of chicken ovalbumin upstream promoter transcription factor II (CoupTF II) ($p < 0.005$, $n = 14$), which tends to clearly co-express either both positively or both negatively with NPY ($p < 0.005$, $n = 18$).

Confirming a well-known relationship, of 45 cell types with extreme values of input resistance ($R_{in}$) and membrane time constant ($\tau_m$) ($p < 0.001$, $n = 45$), 43 are positively correlated ($R^2 = 0.70$). Likewise, high values of the spike width ($AP_{width}$) are associated with high values of $R_{in}$ ($p < 0.001$, $n = 45$), and vice versa for the low values of both ($R^2 = 0.36$).

Of the 69 cell types with axons in more than one layer, only DG Granule and CA1 O-LMR do not have overlapping axons and dendrites ($p < 0.001$, $n = 122$). All of the 43 cell types that have axons in three or more layers have axons that overlap with their dendrites ($p < 0.001$, $n = 122$). Furthermore, cell types with axons in three or more layers tend to have very high $R_{in}$ ($p < 0.001$, $n = 58$) and $\tau_m$ ($p < 0.001$, $n = 59$).

Clear expression of cannabinoid receptor 1 (CB1) implies (in 5/5 cases) clear expression of cholecystokinin (CCK) (if known), and clear absence of CCK implies (in 8/8 cases) clear absence of CB1 (if known) ($p < 0.005$, $n = 15$). Of the two neuron types expressing CCK and not CB1, one expresses VIP; however, independent of CCK expression, CB1 and VIP are not co-expressed in any of the eight types that are positive for either biomarker (NS, $n = 17$). Overall, these patterns suggest that CCK+/VIP+ and CCK+/CB1+ neurons constitute completely separate groups.

The p values and sample sizes (n) pertain to Bernard's exact test on 2 × 2 contingency tables (see 'Materials and methods').

31 entorhinal neuron types, only Oblique Pyramidal cells in LII of medial EC have no layer overlap between axons and dendrites.

Many observed trends also reinforce expected associations and serve as validation for the dataset. For example, 96% (43/45) of neuron types with high or low input resistance have correspondingly high or low $\tau_m$, consistent with RC-circuit theory. The two outliers correspond to data from a paper on CA1 Radiatum Giant cells (*Kirson and Yaari, 2000*), for which the particularly high $\tau_m/R_{in}$ ratio is explained by the oversized soma and consequently large capacitance.

### Potential connectivity

Hippocampome.org contains known connectivity information (synapses or lack thereof, established e.g., by electron microscopy or paired electrophysiological recordings) among its neuron types. Such data, however, are only available for 202 out of 14,884 possible pairs (<2%). In the remaining cases, co-existence of axons and dendrites across parcels allows inference of 'potential connectivity'

between cell types (*Braitenberg, 1991*). Specifically, potential connectivity is computed as the dot-product of the 26-dimensional binary vectors encoding the presence/absence of axons for one neuron and dendrites for another across hippocampal parcels. The possibility of connections can be tentatively excluded for 11,668 pairs of neuron types (78.4%) based on non-overlapping distributions of their respective axons and dendrites. However, incomplete axons and the binary thresholding of the morphological encoding (see 'Materials and methods') might yield false negatives. In 3014 pairs (20%) the axons and dendrites of the two neuron types share at least one parcel. Although the connection probability may not be estimated in the absence of experimental evidence, even the opportunity to make synapses is computationally relevant due to the superior structural plasticity of the hippocampal formation throughout the lifespan (*Leuner and Gould, 2010*).

All (known and potential) connectivity data are summarized in a matrix with rows as pre-synaptic and columns as post-synaptic neuron types (100 neuron types in *Figure 6A*; complete data: hippocampome.org/connectivity). Filled boxes along the main diagonal, corresponding to neurons with axons and dendrites co-located in any parcel (purple boxes in *Figure 1F*), indicate within-type connectivity (not necessarily autapses), and are more frequent among excitatory (87%) than inhibitory (67%) types.

Neuron type connectivity as depicted in *Figure 6A* constitutes an intermediate level of description between the neuron-by-neuron connectome ('synaptome') and the region-by-region connectome ('projectome'). In order for two regions to be connected, there must be at least one neuron type in one region connecting to one neuron type in the other (*Martone and Ascoli, 2013*). Within the hippocampal formation, this relationship is illustrated explicitly in *Figure 6B*: all six areas have potential excitatory, and with the exception of Sub and EC, inhibitory connections to at least one other area, most of which are not considered in the canonical trisynaptic loop (cf. *Burns and Young, 2000*; *van Strien et al., 2009*). The only potential monosynaptic connection between DG and CA1 is inhibitory. While this regional wiring diagram appears simpler than the corresponding seminal illustration for the visual cortex (*Felleman and Van Essen, 1991*), the underlying neuron type circuitry is highly complex. In fact, the explicit wiring diagram of all neuron type potential connections is impractical to render on a single page. Selecting only 15 representative neuron types in the DG entails 253 connections among 43 somatic and dendritic compartments (*Figure 7A*). This compartmental representation of network connectivity captures the computationally distinct subunits that emerge from layer-specific axonal targeting, a key component of mesoscopic neuron type circuitry.

The richness of connectivity is also revealed when focusing on a specific interaction between two neuron types. For instance, in a dual recording of a Granule cell and a CA3 Basket cell, circuit dynamics are potentially affected by many contributors (*Figure 7B*). Granule cells can receive input from 26 neuron types (9 excitatory) across DG, CA3, CA1, and EC. CA3 Basket cells can receive input from 31 types (8 excitatory), including Granule cells and other Basket cells. The above numbers include five neuron types that provide possible input to both Granule and CA3 Basket cells. Granule cells potentially send output to 29 neuron types, counting all 13 that could receive information from CA3 Basket cells, including CA3 Basket cells themselves.

## Use cases

Hippocampome.org can deliver valuable information in many day-to-day research scenarios (*Box 2*) and at multiple stages of a neuroscience study (hippocampome.org/usage). They include determining whether a hippocampal neuron with a particular morphological pattern is known, obtaining a list of candidate neuron types based on partial reconstructions, and finding biomarker and electrophysiological properties for most neuron types. In all highlighted use cases, Hippocampome.org delivers the needed information with very few mouse clicks, in a matter of seconds. Even queries that are nearly impossible with literature search engines (e.g., finding an unnamed neuron by its axonal pattern) allow, at Hippocampome.org, straightforward retrieval of the original evidence.

## Discussion

Systematic organization of present knowledge on neuron types might revolutionize neuroscience akin to the impact the Periodic Table of the Elements had on chemistry 150 years ago. Key to this endeavor is the judicious selection of pivotal variables with the most discriminant and explanatory power. Notably, the initial choice may prove to be a useful proxy even if later recognized as unprincipled:

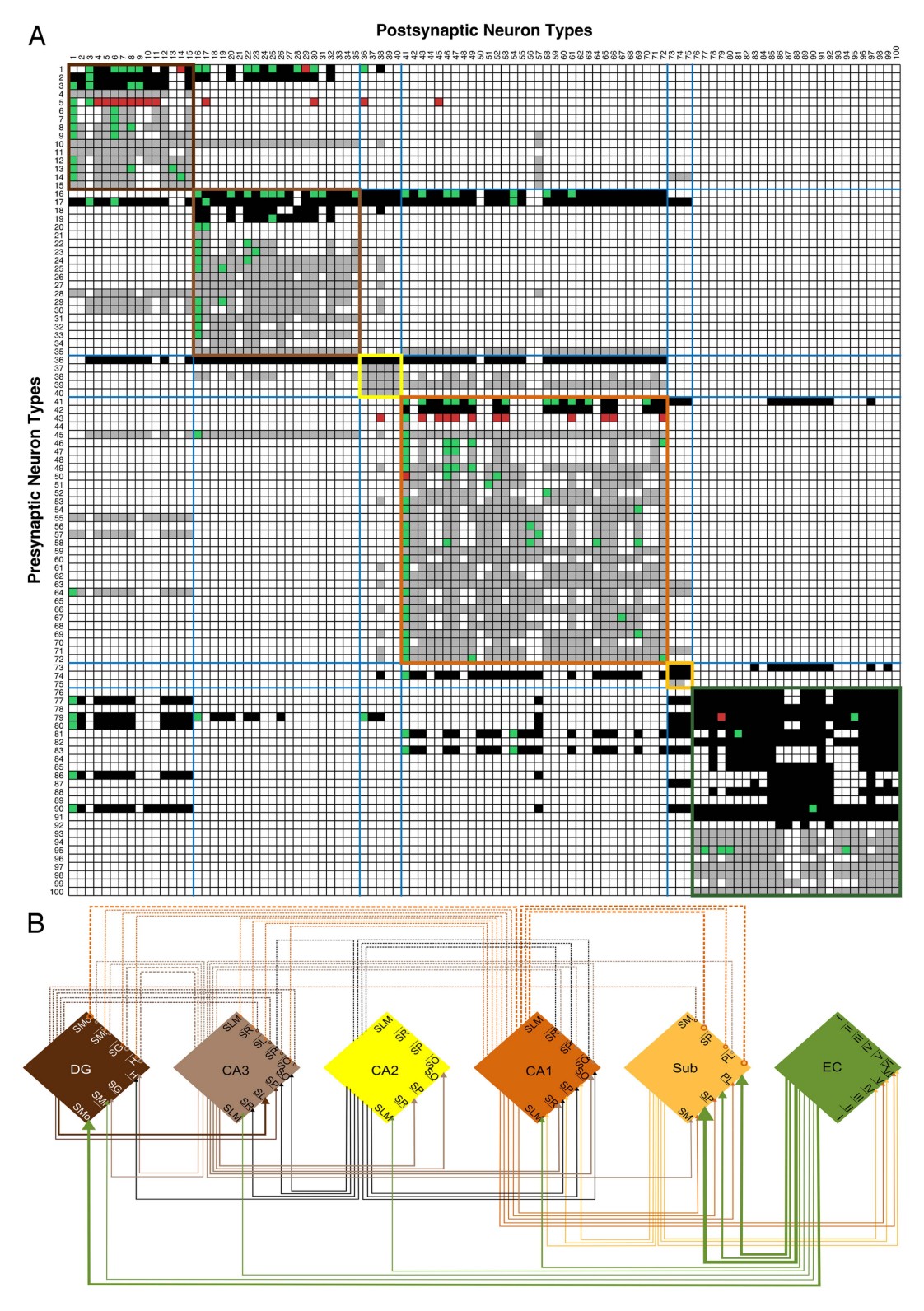

Figure 6. Neuron type connectivity (area color coding and numbering as in *Figure 1*). (**A**) Known and potential connections for 100 neuron types (full matrix: hippocampome.org/connectivity or hippocampome.org/php/images/connectivity/Connectivity_Matrix.jpg), with pre-synaptic types in rows and post-synaptic types in columns. Black squares indicate potential glutamatergic connections and gray squares GABAergic. Pairs of neuron types with experimentally established and refuted synapses are shown using green and red squares, respectively. Colored boundaries demarcate intra-area

*Figure 6. continued on next page*

*Figure 6. Continued*

connections. (**B**) Regional pathways with 33 glutamatergic connections (full lines ending in arrows) and 20 GABAergic connections (dashed lines ending in open circles); line thickness represents the numbers of connected neuron types between areas.

Mendeleev's first attempt to order chemical elements by atomic mass subsequently led to predicting their correct atomic numbers four decades before the discovery of the proton. The distribution of axons and dendrites across anatomical areas and layers is known for the majority of hippocampal neurons identified to date and effectively discriminates among them. Among the features commonly used to describe cortical neurons, morphology is also considerably robust to experimental conditions. Furthermore, just like the positions of the elements in the Periodic Table indicate their potential to combine into molecules, axonal and dendritic patterns provide the blueprint for the potential network connectivity.

Our classification scheme links biochemical, physiological, and synaptic data to structural knowledge of neuron types by methodical literature mining. The cost of this integration is the need to interpret published information to abstract specific evidence from neuron instances into general properties of neuron types. The resulting knowledge base constitutes the first comprehensive machine-readable neuron inventory for a mammalian cortical region. Parallel efforts are underway for the retina (*Siegert et al., 2009*), somatosensory (*Markram, 2006*) and visual cortex (celltypes.brain-map.org), and drosophila (*dos Santos et al., 2015*). Complementary synergies in the rat hippocampal formation include Temporal-lobe.com (*van Strien et al., 2009*), Rat Hippocampus Atlas (*Kjonigsen et al., 2011*), and Hippocampus 3D (*Ropireddy et al., 2012*).

The vision driving Hippocampome.org is a real-scale computer model of the entire hippocampal formation. Accurate simulations require knowledge of the component parts, their locations, numbers, properties, and connectivity. We have started with an accounting of the neuron types defined by their most essential morphological, molecular, and electrophysiological features. Further works-in-progress include neuron counts (*Bezaire and Soltesz, 2013*), firing patterns, synaptic profiles, developmental origins (*Tricoire et al., 2011*), and rhythmic phase-locking (*Somogyi and Klausberger, 2005*). Moreover, the binary assignment of axonal and dendritic distribution across parcels will gradually be substituted by quantitative morphological estimates (*Ropireddy and Ascoli, 2011*).

Neuronal classification will likely benefit by advances in optogenetics, sequencing, and machine learning (*Fenno et al., 2014*; *Kohara et al., 2014*; *Roux et al., 2014*; *Armañanzas and Ascoli, 2015*; *Zeisel et al., 2015*). Rapid growth is expected particularly in the knowledge of molecular markers. Thousands of the ~20,000 genes mapped in the Allen Mouse Brain Atlas (*Lein et al., 2007*) are expressed in the hippocampus. Connecting such large-scale information to the morphology and physiology of particular neuron types will likely answer many open questions while raising new ones.

By integrating available information, Hippocampome.org aims to accelerate discovery. Neuro-Morpho.Org (*Ascoli et al., 2007*), a digital archive of neuronal reconstructions, has enabled numerous secondary findings from amounts of data far outreaching the collection means of individual labs (*Parekh and Ascoli, 2014*). While this article offered selected examples, further in-depth analyses of neuron type connectivity and pairwise property correlations are ongoing.

Consolidation of knowledge also allows assessing the available information to prioritize missing data in the hippocampal formation (cf. *Olshausen and Field, 2005* for V1). The limited interneuron diversity in CA3, CA2, Sub, and EC compared to CA1 suggests several to-be-discovered neurons. Many cell types found in CA1 (e.g., Basket, O-LM, Ivy, Neurogliaform, Axo-axonic, Trilaminar, and Bistratified) have counterparts in some other area(s), but they are probably present in more. As experimental techniques advance, several new neuron types will likely be reported. The Hippo-campome.org framework also sets a standard for the minimal information that must be included in publications describing hippocampal neuron types in order to link the resulting data to the existing body of knowledge.

Initially enforcing strict information granularity has proven critical for the launch of this resource by preventing unending mining of published neuronal properties with clear-cut delimiters. The germinal selection of information to include in Hippocampome.org was pragmatically guided by the density of available published data. For example, in vivo neuronal reconstructions and recordings, although

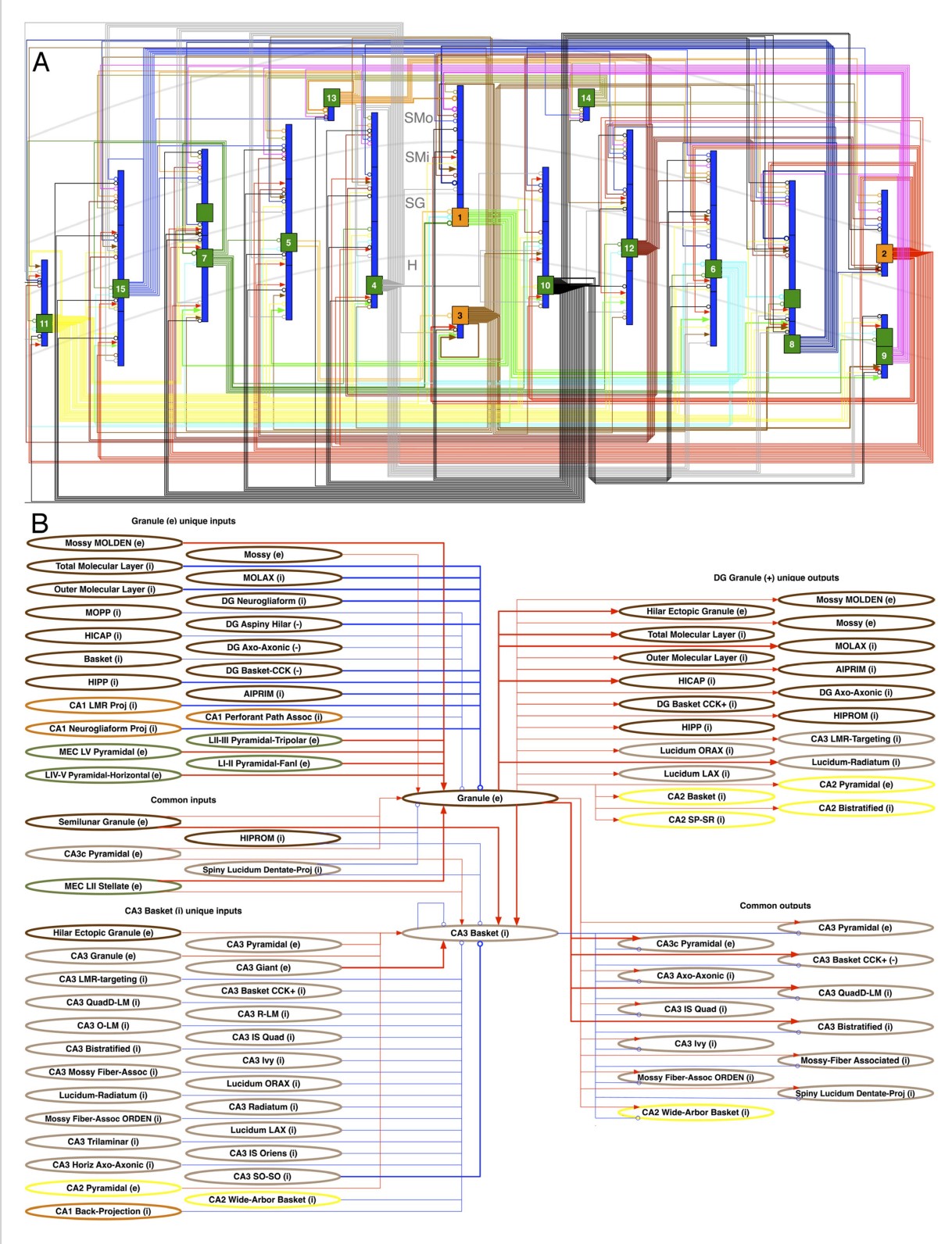

**Figure 7**. Neuron type circuitry. (**A**) Circuit diagram of selected neuron types in DG (full diagram: hippocampome.org/php/images/connectivity/ DG_Circuit_Diagram.jpg or hippocampome.org/php/images/connectivity/DG_Circuit_Diagram.graffle.zip). Axonal connections from pre-synaptic somata (orange glutamatergic, green GABAergic) to post-synaptic somata or dendrites (blue) have unique line colors for each pre-synaptic type for clarity. Lines ending in arrows and open circles indicate, respectively, glutamatergic and GABAergic connections; 22 known (thick lines) and 231 potential (thin)

*Figure 7. Continued*

connections are depicted. 1: Granule. 2: Semilunar Granule. 3: Mossy. 4: AIPRIM. 5: Axo-axonic. 6: Basket. 7: Basket CCK+. 8: HICAP. 9: HIPP. 10: HIPROM. 11: MOCAP. 12: MOLAX. 13: MOPP. 14: Neurogliaform. 15: Outer Molecular Layer. (**B**) Pre- and post-synaptic connections for DG Granule and CA3 Basket cells. Red lines ending in arrows indicate glutamatergic connections, and blue lines ending in open circles indicate GABAergic connections. Thick and thin lines indicate, respectively, known and potential connections. Neuron types are color coded by area of origin (as in *Figure 1*).

extremely valuable, are too sparse to be adopted as defining dimensions in a comprehensive census of neuron types. Similarly, axonal and dendritic patterns are more commonly identified across layers than in the longitudinal (septo-temporal) and transverse (e.g., CA3a,b,c) axes. In limited cases, information is sufficient to distinguish specific neuron types based on finer anatomical sub-divisions. For instance, CA3c Pyramidal cells are differentiated from other CA3 Pyramidal cells, because of their axonal back-projection to DG and lack of dendrites in SLM. Some neuron types are similarly separated between medial and lateral EC.

At the same time, Hippocampome.org is well equipped to manage the expected 'data deluge' from ongoing big science projects (*Kandel et al., 2013*). With accelerating knowledge expansion, our linking protocol to interrelate each additional piece of data with pre-existing information will ensure continuous integration, facilitating user-friendly analysis and modeling. Moreover, Hippocampome. org infrastructure is designed to accommodate additions along many dimensions: new areas (e.g., septum), finer parcels (medial vs lateral EC), additional neuron types (hippocampome.org/ future), and graded reporting of neurite densities. The open-source code of this resource also encourages community-led extension to other brain regions, inter-operability with related initiatives, and progressive adoption of unique identifiers for neuron types and properties.

## Materials and methods

### Anatomical parcels and neuron type identification

Several attempts are underway to establish definitive layer and area boundaries through molecular expression (*Thompson et al., 2008*; *Boccara et al., 2014*). Until formal definitions are in place, generally agreed upon anatomical and histochemical characteristics provide for the delineation of the hippocampal formation into six areas (DG, CA3, CA2, CA1, Sub, and EC) and their respective layers (e.g., CA1 SO, SR, etc.; *Figure 1A,B*). Most of these parcels are standard and can be found in all reviews of the hippocampal formation.

Part of the DG literature only defines a single s. moleculare (SM) between the fissure and SG. Hippocampome.org divides SM into the outer two-thirds (SMo) and the inner one-third (SMi), because SMo receives perforant path input from EC and SMi receives hilar collaterals from mossy cells (*Amaral et al., 2007*).

In Sub, Hippocampome.org lists a single SM superficial to SP and a PL below SP (*van Strien et al., 2009*), differing from the Allen Mouse Brain Atlas (*Lein et al., 2007*) delineation of two parcels above SP and none below.

A neuron type must satisfy four criteria to be included in Hippocampome.org. (1) The soma must be located in the hippocampal formation. (2) The major neurotransmitter must be at least tentatively specifiable; at present, only glutamatergic and GABAergic types have qualified for inclusion in Hippocampome.org (see below for cholinergic). (3) The locations of both axons and dendrites within any of the 26 hippocampal parcels (*Figure 1A,B*) must be clearly presented, either textually or graphically. (4) Experimental evidence of more than a single neuron of the type must be reported.

Hippocampome.org neuron types are primarily distinguished by criterion 3 (axonal–dendritic patterning) independent of author provided names or grouping, but are further differentiated by their synaptic specificity (see *Figure 3*). Neurite densities within parcels, beyond a minimal threshold (see below), are not currently a factor, as most reconstructions are incomplete; this first approximation is likely to change when more quantitative data become available for most neuron types. All evidence collated in Hippocampome.org comes from peer-reviewed articles or book chapters (hippocampome. org/bibliography) and pertains to healthy rodents, predominantly rats and mice ≥13 days old.

## Box 2. Representative Hippocampome.org use case scenarios.

### Property-based recognition of known neuron types

While patching in DG stratum granulosum (SG), a researcher encounters several interneurons with high input resistance ($R_{in}$), membrane time constant ($\tau_m$), and spike amplitude ($AP_{ampl}$). Biocytin filling reveals both axonal and dendritic presences in the inner stratum molecular (SMi) but only axons in the hilus (H). Is this a newly discovered neuron type?

### Hippocampome.org

These characteristics are consistent with those reported for MOCAP neurons, first described by *Markwardt et al. (2011)* as non-Ivy/NG cells.

### Comprehensive listing of potential pre-synaptic sources

In the presence of outward channel blockers, CA1 Pyramidal cells display rebound spikes upon repeated GABA puffing on their distal apical dendrites. What neurons besides O-LM and Neurogliaform cells might trigger such post-inhibitory firing?

### Hippocampome.org

14 different interneurons have axons in stratum lacunosum-moleculare (SLM), including CA1 Quadrilaminar, Back-projection, Radiatum-receiving Apical-targeting, Perforant Path-associated, Oriens-Bistratified Projecting, and LMR cells.

### Comprehensive listing of potential post-synaptic targets

Glutamatergic Cajal–Retzius neurons have recently been observed in the adult rodent hippocampus in larger numbers than previously assumed. What interneurons could they excite in CA1?

### Hippocampome.org

Although the axons of CA1 Cajal–Retzius are confined to SLM, they could nonetheless contact no fewer than 17 types of distinct GABAergic cells, such as Basket, Axo-axonic, Radial Trilaminar, Oriens/Alveus, Schaffer Collateral-associated, as well as several interneuron-specific interneurons.

### Discrimination of perisomatic neuron types by electrophysiological measures

CA3 Basket and Axo-axonic neurons have similar somatic locations, dendritic tree shapes (invading all layers from stratum oriens (SO) to SLM), and overall axonal patterns. Short of the labor-intensive determination of their post-synaptic targets, could they be tentatively distinguished by patch-clamp recording?

### Hippocampome.org

While CA3 Axo-axonic and Basket cells have practically indistinguishable resting potential ($V_{rest}$), $\tau_m$, and $AP_{ampl}$, the former tend to display lower firing threshold ($V_{thresh}$) as well as greater slow afterhyperpolarizations (sAHP) and $R_{in}$.

### Positive identification of neuronal phenotypes

Two varieties of CA2 basket cells have been described: a classic type with dendrites confined within CA2 and a 'wide-arbor' type, whose dendrites enter CA1 and CA3. How can the borders between CA2 and adjacent subregions be reliably demarcated?

### Hippocampome.org

CA2 pyramidal neurons, unlike their CA1 and CA3 counterparts, are positive for Purkinje cell protein 4 (PCP4). Conversely, pyramidal neurons in CA1 and CA3 express α-mannosidase-1

(Man1a), while those in CA2 do not. Immunostaining with either of these markers can thus delineate CA2 boundaries.

**Constraining simulation parameters**

To build a circuit model of grid cell activity, a computational neuroscientist is searching for plausible values for $V_{rest}$, $V_{thresh}$, fast afterhyperpolarizations (fAHP), and $AP_{ampl}$ for the principal cells and main interneurons above the lamina dissecans (LIV) of the medial entorhinal cortex (MEC).

**Hippocampome.org**

These data are available for MEC LII Stellate cells and EC LIII Pyramidal cells (the two major glutamatergic neurons), for EC LII Basket-Multipolar and MEC LIII Superficial Trilayered interneurons (the major perisomatic and dendritic-targeting GABAergic cells, respectively), as well as for eight additional, if less prominent, neuron types (5 excitatory and 3 inhibitory) of layers I-III.

**Distilling information relevant to specific hypotheses**

A novel theory of hippocampal function requires direct feedback inhibition from CA3 and CA1 to DG Granule cells, contrary to the canonical trisynaptic loop and the common assumption of non-projecting GABAergic cells. Does the literature provide any experimental evidence to support the new assumption?

**Hippocampome.org**

Based on axonal–dendritic overlap, both the Granule cell page and the connectivity matrix indicate as (potential) sources of monosynaptic input one CA3 (Spiny Lucidum) and three CA1 (LMR Projecting, Perforant Path-associated, and Neurogliaform Projecting) neurons. Of these, Perforant Path-associated neurons have already been shown to form synapses onto DG Granule cells (Vida et al., 1998).

See hippocampome.org/usage for an extended example.

When incomplete information prevents inclusion in Hippocampome.org, neuron types are placed 'on hold' (hippocampome.org/on-hold). For example, none of the references providing evidence for cholinergic neurons in CA1 clearly describe the axonal and dendritic arbors of individual neurons. On-hold types also include neurons from animals younger than P13 (e.g., DG Cajal–Retzius cells).

## Encoding of neuronal morphology

Hippocampome.org binarizes the locations of axons and dendrites in the 26 parcels: neurites either have 'sufficient' presence in a given parcel or they do not. Lexical and visual thresholds, briefly described here, specify sufficiency criteria (full explanation and examples: hippocampome.org/full-interp). For neuronal reconstructions and schematics, a layer must contain either ≥15% of the overall arbor or at least half the amount included in the most abundant layer. Moreover, the neurite must penetrate ≥15% of the layer depth. In particular, the axonal tree of the CA1 basket cell can spill over from SP into SO and SR, but the penetration threshold is not crossed (cf. Figure 3E from *Pawelzik et al., 2002*). Thus, if the layers are not delineated, the figure may be unusable for neuron typing.

For text descriptions, categorical statements such as 'The dendrites (of CA1 Horizontal Basket cells) are restricted to stratum oriens' (*Maccaferri et al., 2000*), are straightforward. However, quotes that are ambiguous in terms of neurite quantity within a parcel require an interpretive threshold. The term 'most' referring to branches is considered as evidence for presence within that parcel, while the term 'some' is not, although the quote is still included in Hippocampome.org as evidential information. Interpretation of other equivocal statements is carefully annotated.

Although the authors of an article may group a set of neurons together, differing axonal and dendritic patterns can lead to multiple Hippocampome neuron types. This occurs if the number of neurons in each subset is greater than the square root of the total number of neurons described (e.g., if 4 neurons out of 15 have distinct neurite distributions, they would be split into a different cell type because $4 > \sqrt{15}$). For instance, DG neurogliaform cells (*Armstrong et al., 2011*) are divided into two separate neuron types: those that extend their axons into Sub (keeping the name 'DG Neurogliaform') and those that remain local within DG SMo. This latter type is merged with MOPP cells, which have the same axo-dendritic pattern in DG and no projection. Future accumulation of neurite density, molecular expression, and electrophysiology data may result in separating local DG neurogliaform and MOPP cells.

Conversely, neurons assigned to different groups in a paper might belong to the same Hippocampome type, such as R-LM and P-LM cells (*Oliva et al., 2000*), that differing only in somatic location, are merged into a single CA1 OR-LM type in Hippocampome.org.

## Nomenclature

Hippocampome.org neuron names were assigned progressing from the most to the least prominent types. When a single name is used in the literature, that name is adopted into the Hippocampome.org neuron name. For example, DG granule cells in the knowledge base are referred to as DG (e)2201p-CA3_00110 Granule, a formal name encoding the pattern of axons and dendrites (hippocampome. org/brief-interp), and Granule, a common name.

The formal name is composed of multiple parts (hippocampome.org/formal-name), the first of which (e.g., DG (e)2201p) encodes information for the area where the neuron resides: DG designates the home area of the soma; (e) informs about the putative major neurotransmitter being glutamategric; 2201 is the axo–dendritic pattern in the home area ordered from the most superficial to the deepest layer (e.g., dendrites in SMo and SMi, no neurites in SG, and axons in H); and the p indicates that the axons and/or dendrites project out of the home area. The second part of the formal name encoding (e.g., CA3_00110) describes the pattern of the projecting neurites (e.g., axons in SL and SP of CA3). The final portion of the formal name is the same as the common name.

Common neuron names in Hippocampome.org are assigned a prefix denoting their hippocampal area if the same name is used for neurons in different areas (e.g., CA3 Bistratified and CA1 Bistratified), but not if the name is unique (as in the case of HIPP cells, which only exist in DG). Furthermore, the prefix MEC is assigned to neurons that have been characterized solely (or are known to exist predominantly) in the medial EC, such as MEC Layer II Stellate cells. Similarly, the prefix LEC is assigned to lateral entorhinal neurons. In contrast, neurons that are believed to exist in both the medial and lateral ECs are generically referred to with the prefix EC.

When multiple names appear in the literature for the same neuron type, the most frequent or best-known is selected, such as CA1 O-LM over CA1 oriens interneurone of the second type (*McBain et al., 1994*) or CA1 horizontal oriens-alveus interneurone (*Ali and Thomson, 1998*). If no literature name clearly emerges, or if all suitable names are already adopted for more prominent neuron types, we adopt or combine multiple author-originated names, as in EC LI-II Pyramidal-Fan (*Germroth et al., 1991*; *Lingenhöhl and Finch, 1991*; *Empson et al., 1995*; *Tahvildari and Alonso, 2005*). All terms used in publications are always reported as synonyms in Hippocampome.org.

## Linking molecular and electrophysiological data to neuronal morphology

To search for molecular and electrophysiological data, the articles defining the axonal–dendritic pattern of each Hippocampome.org neuron type are mined first. All references citing (webofknowledge.com) or cited by these sources are mined next. Lastly, full-text searches using all known synonyms of the target property and neuron type are performed (scholar.google.com and previously scirus.com). If these searches fail to return usable molecular or electrophysiological information for a given neuron type, the corresponding property is labeled as 'currently unknown' (gray boxes in *Figure 2* and empty entries in *Table 1*).

For inclusion in the knowledge base, molecular and electrophysiological data must be linked to a morphologically defined neuron type. As with the establishment of morphological types, the linking process is blind to neuron names used by authors. Rather, links require either the co-presentation of axonal–dendritic information or the citation, for that specific evidence, of a source that has a morphological description of the neuron. Only in two cases can linking be achieved without complete knowledge of the morphology. The first applies to principal cells within their layers (i.e., granule cells

in DG and pyramidal cells in CA1 SP). These neuron types are readily identified as positive or negative for a biomarker because the layers are >90% homogeneous for the principal cells (*Czéh et al., 2013*) and the somata are densely packed. In the second case, certain groups of neurons can be identified by their axonal tracts, for example, granule cells by their mossy fibers and medial EC LII stellate cells by their perforant path projection. In all cases, the linking information is explicitly included with the evidence reported at Hippocampome.org.

## Molecular biomarker expression

The 20 most studied biomarkers in hippocampal research (*Figure 2*; extended listing: hippo-campome.org/markers) were targeted for literature searches across all 122 neuron types. These include calcium-binding proteins (PV, CB, CR), receptors and transporters (CB1, sub P rec, muscarinic receptor 2, serotonin receptor 3, vesicular glutamate transporter 3, metabotropic glutamate receptor 1α, GABA$_A$-α1 subunit), neuropeptides (CCK, SOM, enkephalin, NPY, VIP, neurogranin), and a miscellaneous group of cytoskeletal and extracellular matrix proteins (α-actinin 2, reelin), transcription factors (COUP-TFII), and enzymes (nNOS).

When information is available, a particular neuron type is characterized as positive, negative, or mixed positive-negative for a biomarker. In the case of mixed expression, the data are evaluated to determine whether the mixed information might be attributed to differences in species (e.g., rats vs mice), techniques (e.g., protein detection vs mRNA detection), or subcellular expression localization (soma vs neurite). When a single population of neurons is shown to be divisible into clearly negative and clearly positive subpopulations, this is taken as an indication of biomarker subtypes. When we are unable to determine whether mixed biomarker data is attributable to species/technique/subcellular localization differences or subtypes, the data are annotated as unresolved.

## Electrophysiological properties

Electrophysiological property values are extracted from the literature and compiled, when available, for all Hippocampome.org neuron types. The knowledge base includes passive ($R_{in}$, $\tau_m$, resting membrane potential or $V_{rest}$), spike ($AP_{ampl}$, $AP_{width}$, $V_{thresh}$, fAHP), and other response parameters (maxFR, sAHP, hyperpolarization sag ratio). Values are extracted from published reports either from text (or tables) or by digitizing figures (plotdigitizer.sourceforge.net) and reported as mean, range, standard deviation, and number of measurements. Although these properties can be defined and measured in multiple ways, Hippocampome.org standardizes the data according to a single definition (hippocampome.org/ephys-defs). This standardization, along with full hand-curation, distinguishes our approach from the semi-automated mining by NeuroElectro (*Tripathy et al., 2014*).

## Maintenance and further development

The growth, evolution, and accuracy of Hippocampone.org content rely on two main mechanisms: continuous literature mining (hippocampome.org/ongoing-mining) and community feedback (hippo-campome.org/feedback). We update the bibliographic listing at quarterly intervals (based on citation alerts for many of the core review articles as well as perusals of new issues of most mainline neuroscience publications), adding new relevant references and linking to the knowledge base the articles that have been annotated. Moreover, we welcome suggestions for improvements, corrections, and additions. Addressing of this feedback will also be incorporated into the Frequently Asked Questions (FAQs) listing (hippocampome.org/FAQ) for future reference by all users. In order to ensure reliable resource citation, we adopt a numbered versioning system to release additions of new neuron types and specification of additional properties for existing neuron types. Publication of this article marks the v.1.0 release of Hippocampome.org.

## Web portal and database

The web portal and associated database infrastructures facilitate access to and utilization of morphological, molecular, electrophysiological, and connectivity information. The implementation leverages the model-view-control software design. The model component, which defines the database interface, is provided solely by server-side code. The view component, which renders the web pages, and the control code, which implements decision logic, are both served up by the server,

but are run in the user's browser. The underlying relational database ensures flexibility in establishing relations between data records.

Hippocampome.org is deployed on a CentOS 5.11 server running Apache 2.2.22 and runs on current versions of Firefox, Chrome, Safari, and Explorer. Knowledge base content is served up to the PHP 5.3.27 website from a MySQL 5.1.73 database. Django 1.7.1 and Python 3.4.2 provide database ingest capability of comma separated value annotation files derived from human-interpreted peer-reviewed literature. Hippocampome.org code is available open source at github.com/Hippo-campome.org.

## Pairwise correlation analysis

We explored pairwise correlations between 205 properties of Hippocampome neuron types, including neurotransmitter; axonal, dendritic, and somatic locations in the 26 partitions and 6 areas; the projecting (inter-areas) or local (intra-area) nature of axons and dendrites; axon and dendrite co-presence within any partition; axonal and dendritic presence in a single layer only or in $\geq$3 layers; clear positive or negative expression of any biomarkers; and high (top third) or low (bottom third) values for seven electrophysiological properties (excluding highly stimulus-dependent sag ratio, sAHP, and maxFR). To evaluate the correlations between these categorical properties, we use $2 \times 2$ contingency matrices with Barnard's exact test, which provides the greatest statistical power when row and column totals are free to vary (*Lydersen et al., 2009*).

## Acknowledgements

We thank Maurizio Bergamino, Faramarz Faghihi, Ben Holmes, Sean Mackesey, Keivan Moradi, Mona Suliman, Siva Venkadesh, and many student interns (hippocampome.org/thx) for invaluable help. This project is supported by grants R01NS39600 (NIH), MURI N00014-10-1-0198 (ONR), Nakfi (Keck), CENTEC (AFOSR), Robust Intelligence (NSF), and Northrop Grumman.

## Additional information

### Funding

| Funder | Grant reference | Author |
| --- | --- | --- |
| National Institutes of Health (NIH) | R01NS39600 | Giorgio A Ascoli |
| Office of Naval Research (ONR) | MURI N00014-10-1-0198 | Giorgio A Ascoli |
| Air Force Office of Scientific Research (AFOSR) | CENTEC | Giorgio A Ascoli |
| National Science Foundation (NSF) | Robust Intelligence | Giorgio A Ascoli |
| Northrop Grumman | | Giorgio A Ascoli |
| Keck | Nakfi | Giorgio A Ascoli |

The funders had no role in study design, data collection and interpretation, or the decision to submit the work for publication.

### Author contributions

DWW, CMW, CLR, AOK, GAA, Conception and design, Acquisition of data, Analysis and interpretation of data, Drafting or revising the article; DJH, Conception and design, Analysis and interpretation of data, Drafting or revising the article

### Author ORCIDs

Diek W Wheeler, http://orcid.org/0000-0001-8635-0033
David J Hamilton, http://orcid.org/0000-0001-6265-9917

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
