## [Decision Letter]

Thank you for submitting your work entitled “Hippocampome.org: A knowledge base of neuron types in the rodent hippocampus” for peer review at *eLife*. Your submission has been favorably evaluated by Eve Marder (Senior Editor) and three reviewers, one of whom is a member of our Board of Reviewing Editors.

The reviewers have discussed the reviews with one another and the Reviewing Editor has drafted this decision to help you prepare a revised submission.

All of the reviewers were in agreement that Hippocampome.org potentially represents an excellent resource for the community. While several improvements and suggestions were made, it was recognized that it was not essential that they all be addressed immediately as there are many. The authors are urged to consider the Research Advance option of *eLife* for future Hippocampome.org developments.

To allow the authors to access the full range of suggestions that arose, they are appended below. In summary, the essential revision requirements are as follows:

1) A better and clearer presentation in the paper is needed. Specifically, some re-organization with Methods/Results needs to be considered – a flowchart of design criteria/strategy might be helpful, and several figures need improvement.

2) A well-described plan for the maintenance, curation and updating of hippocampome.org is needed.

Reviewer #1:

We would like to applaud and thank the authors for undertaking this endeavour – it should be helpful in bringing experimental and modeling communities closer together and to shorten literature exploration time.

Hippocampome.org represents a potentially helpful resource for the large hippocampus community of experimentalists and modellers. To get it started, we think the approach taken of not being overly restrictive but at the same time having some sort of design criteria is a good one. The authors have done extensive literature data mining to organize the available information for the hippocampus community. Our overall assessment is that this could become a ‘go-to’ resource for many people. However, we emphasize “potentially helpful” as we identified various aspects that we thought required improvement before being ready for public consumption.

In reviewing the manuscript, we also consulted the online hippocampome.org. In general, there could be better ‘coordination’ between the manuscript description and the online version. See comments below:

1) There should be a summary description of the design criteria used on the website, as coordinated with the manuscript. Presumably, an accepted manuscript will be linked there, but if I did not have the manuscript in hand, many more questions would've arisen as to choices made in choosing cell types etc.

In essence, there needs to be much more explanation and information up front. Perhaps a simple flow chart of criteria could be presented up front (on the website and as a paper figure?), and details (special cases/exceptions) given in 'help' via clicking (to specify on front page). There seems to be general criteria (cell types distinguished via axon/dendrites in particular layers), and then additional special cases and exceptions (e.g. BC and AAC etc, biomarkers etc). This is understandable given the current state of what is known and what has been done, but there will be varying knowledge in the community and as such, various assumptions made, and without design criteria clearly set down, the resource could lead to confusion rather than being helpful, which would be unfortunate. In general, much more clarity as to how decisions for distinguishing neuron types were made is needed.

Within the manuscript itself, Results and Methods seemed mixed in presenting general and specific criteria – better organization is required, and perhaps using the flowchart idea of general and specific criteria would be helpful in organizing the manuscript presentation? For example, biomarkers mentioned in Results (but not Methods) whereas anatomy/morphology mentioned in both. It is not clear what the authors intend ‘Methods’ and ‘Results’ to encompass. They could provide some explanations and describe how they have chosen to organize the manuscript.

While we think this will be an excellent and helpful resource, the lack of clarity on how choices were made could lead to confusion for people new to the hippocampal field as well as disputes/different opinions about why and how choices were made. Along with full explanations for design choices, the rationale for choices could also be presented together (e.g. rat data for electrophysiological is presumably simply because more data is available?), and whether people agree or not about different details would be a moot point, as it would be clearly set down in black and white so that results from using Hippocampome.org could be interpreted from the design criteria if clearly presented.

2) Given the amount of work involved to maintain and update the database, it would probably be a good idea to get the community to help keep the database updated (as well as identifying errors – we think we found a couple?), but with curation. Presumably the authors have thought about this? Their suggestions could be provided in the manuscript. Referring back to our first comment, clear design criteria explanations would have to be provided.

Reviewer #2:

The authors introduce and describe their online database and corresponding webtool, Hippocampome.org. Hippocampome.org is a well-organized collation of data gleaned from the body of published hippocampal research, whose scope currently includes morphology, cell location, neurotransmitters released, markers expressed, and cell electrophysiology. This tool represents an impressive amount of work and thought. It will be a valuable resource for experimentalists, theoreticians, and modelers. It also makes the existing body of published neuroscience data much more useful and accessible. This work is of potentially great significance, and is a good fit for a journal such as *eLife*. I recommend acceptance pending the resolution of the points below:

1) Who is doing the annotating and review? More information about how often the literature is combed, who is extracting the data and who is reviewing the entries made in the database, ensuring completeness and quality control, would inspire confidence in the reader. Although the description of how the literature is surveyed is quite helpful, it would also be helpful to elaborate on the frequency of the survey activities (daily, weekly, etc.) and any mechanisms for Hippocampome.org users to suggest the inclusion/review of specific articles.

2) I recommend briefly discussing and citing NeuroElectro (http://www.neuroelectro.org/publications/), as it appears to overlap somewhat with the ephys portion of Hippocampome. It may also be the case that Hippocampome can share mined data with NeuroElectro, or vice versa (though NeuroElectro does not standardize their ephys calculations as Hippocampome does).

3) Is there a mechanism for discussion/resolution of conflicting groupings or definitions of neuron types? A brief discussion of any obstacles to the broader community adopting the conventions and standards for neuron grouping presented here could be useful if there is room in the text.

4) Certainly some standard must be set to classify the layers of a cell type's neurites, and >15% sounds logical. However, that same cutoff should be remembered when discussing the “tentative” impossibility of connections between certain cell types due to non-overlapping layers. I recommend rewording the first paragraph of the subsecton “Potential connectivity” which refers to “complete segregation”.

Reviewer #3:

The authors report on work in progress on a freely accessible web-based searchable database of hippocampal neuronal types based primarily on axonal and dendritic layer location (as well as more detailed connectivity information when available), neurotransmitter phenotype, molecular biomarkers and electrophysiology. The database is ready-to-use, but is flexible enough to be continuously updated and expanded. This is an important development in the field, partly because it enables search beyond the standard literature bases, and partly because it sets a standard for the minimal information that must be included in papers describing neuronal types for such papers to be useful to others, and, moreover, because it makes it evident what information would be desirable for such a database to become quantitative in future. The development of such database may be as important for the understanding of circuit functions in the brain as the genome wide atlases of neuronal gene expression are for single cell properties (e.g. Lein et al., Nature 2007), and the current data base strikes me as a good start and a very useful resource for the hippocampal community.

As a potential user of this database, I found it very attractive for several reasons.

First, I like the emphasis on information relevant for the synaptic connectivity, partly because this is important for understanding circuit functions and building network models, and partly because these are relatively reliable categorical variables, enabling the development of a robust database. Second, I like the attempt to make the cell type terminology generic and independent of the names used by the original authors. Third, I like the links to the original literature, probably one of the most useful features of this data base. Fourth, I like the open access and open-source code, enabling other investigators to build on this work.

I understand the necessity of using categorical variables at this early stage of development, but in future, continuous variables should be used, in order to facilitate quantitative modelling. The authors allude to the possibility that this will be incorporated in due time.

The paper is in general well presented. However, I had problems with some of the figures:

1) In Figure 3, it was unclear to me what the Balloon plot in Figure 3 actually is intending to show. More explanation in the figure legend might help, but better still would be the replacement by a simpler figure to illustrate the point to be made.

2) Figure 5 illustrates the numbers of connected neuron types between areas. I am not sure how interesting such a metric is, as it is highly dependent on how you define neuronal type. Equally or more interesting for any modeller would be the density of each of these connections, if this information is available in the database. Alternatively, this figure could probably be left out altogether.

3) Figure 6 is difficult to read and does not really tell me more than the complexity of the connectivity. I am wondering whether this type of diagram is useful at all for most readers, unless it is required to understand Figure 6, but I could not find any such link described.

4) In Figure 1, it is confusing that SMo is labelled differently in the left and right panel.

5) In Table 3, reference is made to MEC LII stellate cells but EC LIII pyramidal cells [without the M], and EC LII basket-multipolar [without the M] but MEC LIII superficial trilayered interneurons. Is this intentional?

6) Some abbreviations appear not to be explained. For example in the fourth paragraph of the subsection “Identifying neuron types by axonal and dendritic patterns”, what does R-targeting mean – is it SR-targeting (which is defined), or does it mean something different?

[Editors' note: further revisions were requested prior to acceptance, as described below.]

Thank you for submitting your work entitled “Hippocampome.org: A knowledge base of neuron types in the rodent hippocampus” for peer review at *eLife*. Your submission has been favorably evaluated by Eve Marder (Senior Editor) and three reviewers, one of whom is a member of our Board of Reviewing Editors.

The reviewers all felt that their concerns have been adequately addressed in the revised version, and have a few suggested revisions before a final acceptance. The revisions are as follows.

In the paper:

1) Given the nature of the work, it was felt that “Description of Resource” rather than “Results” would be a more appropriate heading. Please adjust accordingly in the revised text. Also, please expand the paper description part at the end of the Introduction to say something like “A Discussion follows the ‘Description of Resource’ and in ‘Methods’ we provide information on…”.

2) The link for the “on hold” types is listed as hippocampome.org/on-hold (third paragraph of subsection “Identifying neuron types by axonal and dendritic patterns”, subsection “Summary of design criteria” and sixth paragraph of subsection “Anatomical parcels and neuron type identification”), however, the actual link is http://hippocampome.org/php/Help_On-hold_Types.php (as per the link from the Help menu).

I think you didn't put an alias for hippocampome.org/on-hold to this other link, as you did for hippocampome.org/full-interp – i.e. it's a technical misstep that you can easily fix.

On the website:

A bit of re-organization of the Help section would be helpful.

1) Move FAQs to the top (rather than buried in Miscellaneous).

2) Move Interpretation Protocols Flowchart (currently at the end of “Hi-resolution images” to the top.

3) Move Formal Name Encoding (currently in “Miscellaneous”) to the top.

Perhaps a “General” section just before or after “Feedback” could be made for these above 3 items.

4) On the front page, put a link to “Feedback” that appears in Help now so that user is alerted to this option.

5) Refer to the *eLife* paper on front page and/or at top of “Help” in the “General” section with the other key aspects (1-3 above).

---

## [Author Response]

Reviewer #1:

1) There should be a summary description of the design criteria used on the website, as coordinated with the manuscript. Presumably, an accepted manuscript will be linked there, but if I did not have the manuscript in hand, many more questions would've arisen as to choices made in choosing cell types etc.

*In essence, there needs to be much more explanation and information up front. Perhaps a simple flow chart of criteria could be presented up front (on the website and as a paper figure?), and details (special cases/exceptions) given in ‘help’* via *clicking (to specify on front page). There seems to be general criteria (cell types distinguished* via *axon/dendrites in particular layers), and then additional special cases and exceptions (e.g. BC and AAC etc, biomarkers etc). This is understandable given the current state of what is known and what has been done, but there will be varying knowledge in the community and as such, various assumptions made, and without design criteria clearly set down, the resource could lead to confusion rather than being helpful, which would be unfortunate. In general, much more clarity as to how decisions for distinguishing neuron types were made is needed.*

We appreciate this criticism. We have inserted a flowchart illustration summarizing the design criteria, as suggested, both in the manuscript (new Figure 3) and in the Hippocampome.org online portal under the high-resolution figures portion of the “Help” section. Moreover, we have added a new “Summary of design criteria” section in the Results.

Within the manuscript itself, Results and Methods seemed mixed in presenting general and specific criteria – better organization is required, and perhaps using the flowchart idea of general and specific criteria would be helpful in organizing the manuscript presentation? For example, biomarkers mentioned in Results (but not Methods) whereas anatomy/morphology mentioned in both. It is not clear what the authors intend ‘Methods’ and ‘Results’ to encompass. They could provide some explanations and describe how they have chosen to organize the manuscript.

Since the manuscript is structured with the Results first and the Methods last, we had to describe the overall methodological approach, at least at a general level, with the Results. Following the reviewer’s recommendation, we have added at the end of the Introduction a paragraph explaining the organization of the manuscript. Moreover, we have moved all non-essential technical details from the Results to the Methods.

*While we think this will be an excellent and helpful resource, the lack of clarity on how choices were made could lead to confusion for people new to the hippocampal field as well as disputes/different opinions about why and how choices were made. Along with full explanations for design choices, the rationale for choices could also be presented together (e.g. rat data for electrophysiological is presumably simply because more data is available?), and whether people agree or not about different details would be a moot point, as it would be clearly set down in black and white so that results from using*
*Hippocampome.org*
*could be interpreted from the design criteria if clearly presented.*

The manuscript now refers to a newly added “Frequently Asked Questions (FAQ)” section in the online portal (hippocampome.org/FAQ) that explains, among other details, the rationale for the Hippocampome.org design choices.

2) Given the amount of work involved to maintain and update the database, it would probably be a good idea to get the community to help keep the database updated (as well as identifying errors – we think we found a couple?), but with curation. Presumably the authors have thought about this? Their suggestions could be provided in the manuscript. Referring back to our first comment, clear design criteria explanations would have to be provided.

We agree with the need to discuss plans to keep the database updated. We have added a “Maintenance and Further Development” section in the Methods that describes mechanisms for community involvement and feedback. While we intend to maintain for the foreseeable future the role of curators to ensure consistent application of the design criteria, we have implemented (and described in the revised manuscript) a web-based form soliciting suggestions, criticisms, requests, and questions from the community (http://hippocampome.org/php/user_feedback_form_entry.php).

Reviewer #2:

*1) Who is doing the annotating and review? More information about how often the literature is combed, who is extracting the data and who is reviewing the entries made in the database, ensuring completeness and quality control, would inspire confidence in the reader. Although the description of how the literature is surveyed is quite helpful, it would also be helpful to elaborate on the frequency of the survey activities (daily, weekly, etc) and any mechanisms for*
*Hippocampome.org*
*users to suggest the inclusion/review of specific articles.*

We have added a “Maintenance and Further Development” section in the Methods that provides the missing details (see also hippocampome.org/php/Help_Morphological_Bibliographic_Protocols.php). This point is also addressed in the FAQ.

*2) I recommend briefly discussing and citing NeuroElectro (**http://www.neuroelectro.org/publications/**), as it appears to overlap somewhat with the ephys portion of Hippocampome. It may also be the case that Hippocampome* can *share mined data with NeuroElectro, or vice versa (though NeuroElectro does not standardize their ephys calculations as Hippocampome does).*

Indeed, NeuroElectro overlaps somewhat with the ephys portion of the Hippocampome, but it does not standardize the parameter calculations. The reviewer’s suggestion is well taken, and in fact Hippocampome.org did share mined data with NeuroElectro. The NeuroElectro curator visited our lab in May 2014 and we provided the Hippocampome.org ephys data as a ‘gold standard’ testbed to help validate the NeuroElectro automated literature mining protocol. We have checked NeuroElectro content related to hippocampus neurons and found no additional information to date, but we will continue to monitor future updates. We have added a brief discussion of and citation to NeuroElectro in the manuscript. We also list NeuroElectro in the “Other Useful Links” of the Hippocampome.org Help section and have added a discussion of the relationship between these two resources in the FAQ.

3) Is there a mechanism for discussion/resolution of conflicting groupings or definitions of neuron types? A brief discussion of any obstacles to the broader community adopting the conventions and standards for neuron grouping presented here could be useful if there is room in the text.

We have added an online form (http://hippocampome.org/php/user_feedback_form_entry.php) to solicit community involvement, and will address relevant posts in the FAQ section. If additional posts refer to previous comments, we record the discussion online in the form of threaded conversation (http://hippocampome.org/php/Help_Feedback_Submissions.php). If/when consensus emerges, it will be incorporated in the knowledge base.

4) Certainly some standard must be set to classify the layers of a cell type's neurites, and >15% sounds logical. However, that same cutoff should be remembered when discussing the “tentative” impossibility of connections between certain cell types due to non-overlapping layers. I recommend rewording the first paragraph of the subsecton “Potential connectivity” which refers to “complete segregation”.

Thank you for pointing this out. The manuscript has been edited as recommended.

Reviewer #3:

The authors report on work in progress on a freely accessible web-based searchable database of hippocampal neuronal types based primarily on axonal and dendritic layer location (as well as more detailed connectivity information when available), neurotransmitter phenotype, molecular biomarkers and electrophysiology. The database is ready-to-use, but is flexible enough to be continuously updated and expanded. This is an important development in the field, partly because it enables search beyond the standard literature bases, and partly because it sets a standard for the minimal information that must be included in papers describing neuronal types for such papers to be useful to others, and, moreover, because it makes it evident what information would be desirable for such a database to become quantitative in future. The development of such database may be as important for the understanding of circuit functions in the brain as the genome wide atlases of neuronal gene expression are for single cell properties (e.g. Lein et al., Nature 2007), and the current data base strikes me as a good start and a very useful resource for the hippocampal community.

We are grateful for the excellent comment on publication minimal information standards, which we have incorporated in the Discussion.

As a potential user of this database, I found it very attractive for several reasons.

First, I like the emphasis on information relevant for the synaptic connectivity, partly because this is important for understanding circuit functions and building network models, and partly because these are relatively reliable categorical variables, enabling the development of a robust database. Second, I like the attempt to make the cell type terminology generic and independent of the names used by the original authors. Third, I like the links to the original literature, probably one of the most useful features of this data base. Fourth, I like the open access and open-source code, enabling other investigators to build on this work.

I understand the necessity of using categorical variables at this early stage of development, but in future, continuous variables should be used, in order to facilitate quantitative modelling. The authors allude to the possibility that this will be incorporated in due time.

We agree with this comment and have added a sentence in this regard to the Discussion.

The paper is in general well presented. However, I had problems with some of the figures:

*1) In*
Figure 3*, it was unclear to me what the Balloon plot in*
Figure 3
*actually is intending to show. More explanation in the figure legend might help, but better still would be the replacement by a simpler figure to illustrate the point to be made.*

The balloon plot (now Figure 4) is meant to show the uneven distribution of data across the knowledge dimensions and hippocampal areas. We have reorganized the section “A digital storehouse of explicit knowledge” of the Results to make the message more explicit and we have added a sentence in the figure legend to help direct more clearly the readers’ attention.

*2)*
Figure 5
*illustrates the numbers of connected neuron types between areas. I am not sure how interesting such a metric is, as it is highly dependent on how you define neuronal type. Equally or more interesting for any modeller would be the density of each of these connections, if this information is available in the database. Alternatively, this figure could probably be left out altogether.*

Unfortunately, the density of the connections is not available in the database. Although the number of connected neuron types between areas depends on the definition of neuron types, so does almost all the content of this paper. At the same time, this metric may be interesting to many readers in that it directly links neuron type circuitry with regional connectivity. We have edited the text to explain this relationship more clearly.

*3)*
Figure 6
*is difficult to read and does not really tell me more than the complexity of the connectivity. I am wondering whether this type of diagram is useful at all for most readers, unless it is required to understand*
Figure 6*, but I could not find any such link described.*

We have edited the text to explicitly compare the complexity of this diagram with Figure 5 and with the seminal 1991 connectome of the visual system by Felleman and Van Essen. In order to make the wiring diagram more useful for other readers, we are now also providing in the legend a link to an editable version of the high-resolution figure source.

*4) In*
Figure 1*, it is confusing that SMo is labelled differently in the left and right panel.*

Thanks. Figure 1 has been edited to correct the inconsistency.

5) In Table 3, reference is made to MEC LII stellate cells but EC LIII pyramidal cells [without the M], and EC LII basket-multipolar [without the M] but MEC LIII superficial trilayered interneurons. Is this intentional?

Yes, it is intentional. We assign the prefix MEC to neurons that have been characterized solely (or are known to exist predominantly) in the Medial Entorhinal Cortex, such as layer II spiny stellate cells. Similarly, the prefix LEC is assigned to lateral entorhinal neurons. In contrast, neurons that are believed to exist in both the medial and lateral entorhinal cortices are given the prefix EC. We have added an explanation in the “Nomenclature” section of the Methods as well as a Q/A in the FAQ addressing this issue.

6) Some abbreviations appear not to be explained. For example in the fourth paragraph of the subsection “Identifying neuron types by axonal and dendritic patterns”, what does R-targeting mean – is it SR-targeting (which is defined), or does it mean something different?

“R-targeting” has now been amended to read “Radiatum-targeting” in the manuscript.

[Editors' note: further revisions were requested prior to acceptance, as described below.]

The reviewers all felt that their concerns have been adequately addressed in the revised version, and have a few suggested revisions before a final acceptance. The revisions are as follows.

In the paper:

1) Given the nature of the work, it was felt that “Description of Resource” rather than “Results” would be a more appropriate heading. Please adjust accordingly in the revised text. Also, please expand the paper description part at the end of the Introduction to say something like “A Discussion follows the ‘Description of Resource’ and in ‘Methods’ we provide information on…”.

We have changed the name of the Results section and added several sentences expanding upon the content of the Methods section.

*2) The link for the “on hold” types is listed as*
*hippocampome.org/on-hold*
*(third paragraph of subsection “Identifying neuron types by axonal and dendritic patterns”, subsection “Summary of design criteria” and sixth paragraph of subsection “Anatomical parcels and neuron type identification”), however, the actual link is*
*http://hippocampome.org/php/Help_On-hold_Types.php*
*(as per the link from the Help menu).*

This has now been corrected.

On the website:

A bit of re-organization of the Help section would be helpful.

1) Move FAQs to the top (rather than buried in Miscellaneous).

2) Move Interpretation Protocols Flowchart (currently at the end of “Hi-resolution images” to the top.

3) Move Formal Name Encoding (currently in “Miscellaneous”) to the top.

Perhaps a “General” section just before or after “Feedback” could be made for these above 3 items.

These three items have now been moved to a new “General” section that is listed just before the “Feedback” section on the Help page.

4) On front page, put a link to “Feedback” that appears in Help now so that user is alerted to this option.

Mention of the feedback form has now been added to the main front page.

*5) Refer to the* eLife *paper on front page and/or at top of “Help” in the “General” section with the other key aspects (1-3 above).*

Reference to the current manuscript has been added to the front page. After formal acceptance, a link to the manuscript will also be added.